# Equilibrium and non-Equilibrium regimes in the learning of Restricted Boltzmann Machines

**Aurélien Decelle**[1,2,†]        **Cyril Furtlehner**[2]

**Beatriz Seoane**[1,⋆]

[1]Departamento de Física Teórica, Universidad Complutense, 28040 Madrid, Spain.
[2]Université Paris-Saclay, CNRS, INRIA Tau team, LISN, 91190, Gif-sur-Yvette, France.
[†]adecelle@ucm.es; ⋆bseoane@ucm.es

## Abstract

Training Restricted Boltzmann Machines (RBMs) has been challenging for a long time due to the difficulty of computing precisely the log-likelihood gradient. Over the past decades, many works have proposed more or less successful training recipes but without studying the crucial quantity of the problem: the mixing time, i.e. the number of Monte Carlo iterations needed to sample new configurations from a model. In this work, we show that this mixing time plays a crucial role in the dynamics and stability of the trained model, and that RBMs operate in two well-defined regimes, namely equilibrium and out-of-equilibrium, depending on the interplay between this mixing time of the model and the number of steps, $k$, used to approximate the gradient. We further show empirically that this mixing time increases with the learning, which often implies a transition from one regime to another as soon as $k$ becomes smaller than this time. In particular, we show that using the popular $k$ (persistent) contrastive divergence approaches, with $k$ small, the dynamics of the learned model are extremely slow and often dominated by strong out-of-equilibrium effects. On the contrary, RBMs trained in equilibrium display faster dynamics, and a smooth convergence to dataset-like configurations during the sampling. Finally we discuss how to exploit in practice both regimes depending on the task one aims to fulfill: (i) short $k$ can be used to generate convincing samples in short learning times, (ii) large $k$ (or increasingly large) is needed to learn the correct equilibrium distribution of the RBM. Finally, the existence of these two operational regimes seems to be a general property of energy based models trained via likelihood maximization.

## 1   Introduction

Restricted Boltzmann Machines (RBM) are one of the generative stochastic neural networks that have been available for decades [1] and are renowned for their capacity to learn complex datasets [2] and draw similar new data. Furthermore, the hidden correlations found in the data are easily accessible [3, 4, 5], which is particularly interesting for a scientific use of Big Data. Despite these positive features, RBMs still remain hard to train and evaluate, and hence to use in practical problems. In comparison, Generative Adversarial Network (GAN) [6] (for which it is much harder to find a working architecture) are more commonly used, not only because of the advantages given by convolutional layers, but also because the generative process does not depend on a costly sampling procedure with an uncontrolled convergence time.

35th Conference on Neural Information Processing Systems (NeurIPS 2021).

Both training and data generation of RBMs are tricky and unpredictable in inexperienced hands. One of the main difficulties is that it is hard to evaluate if the learning is progressing or not. Yet, many works in the past twenty years have proposed recipes for good practices in RBM training [7]. Unfortunately, the evaluation of these recipes relies only on the comparison of the values reached by the log-likelihood (LL) [8, 7, 9], a quantity that cannot be monitored in tractable times during the learning process. Studies properly characterizing the quality, independence, or stability of the generated samples using the different recipes are nearly absent in the Literature. In fact, many works just show a set of new samples (for eye evaluation) that were either obtained after a short Markov Chain Monte Carlo (MCMC) sampling initialized at the dataset, or borrowed from the permanent chain (negative particle) at the end of the training [10]. In other studies, a reconstruction error after one or several MCMC steps is used to determined if the machine was correctly trained [11]. However, none of these measures guarantees that the trained model can generate from scratch new samples nor that the dataset are typical samples of that model. At most these tests assure you that samples of the dataset are locally stable w.r.t few MCMC steps, unless a proper decorrelation from the initial conditions or a link to equilibrium properties are investigated. Finally, more recent works use a classification score on the activation of the latent variables [12, 13], which reflects that the relevant features were learned, but is silent about the generative power of the machine.

RBMs are receiving an increasing attention in life and pure sciences during the recent years [5, 14, 15, 16, 17, 18, 19, 20, 21] given their potential for interpretability of the patterns learned by the machine (a rare feature in machine learning). Indeed, the RBM formalism enables the extraction of the (unknown) probability distribution function of the dataset. Yet, such an approach is meaningful if the dataset is related to the equilibrium properties of the trained model. Given these new and exciting prospectives for the RBMs, it is more than ever important to establish reproducible protocols and evaluation tools to guarantee not only that the model generates good enough data, but also that the dataset are typical equilibrium samples of the model.

In this work we demonstrate that the classical training procedures can lead to two distinct regimes of the RBM: an equilibrium and an out-of-equilibrium (OOE) one. A finding that had been recently discussed in the Literature under the name of convergent or non-convergent regimes in a different family of generative energy based models (EBMs), those whose energy function is a ConvNet [22]. Here, at variance to previous approaches, we show that the equilibrium regime is observed if the number of MCMC steps, $k$, used to estimate the LL gradient, is larger than the algorithm mixing time (MT). The latter emerges when $k$ falls below this MT. In particular, we show that for a classical dataset such as MNIST, the MT is always rather large and grows with the learning time. This implies that learning an "equilibrium" model (i.e. a RBM that operates in the equilibrium regime) is costly, and that most of the works in the Literature operate without any doubt in the OOE regime. In this regime, the equilibrium distribution of the machine at the end of the learning is significantly different from that of the dataset. Yet, despite this handicap, we show the OOE regime can be exploited to generate good samples at short training times, an observation that have already been exploited in practice [23] for convolution EBM. All our conclusions rely on the study of 9 datasets: MNIST [24], whose results are discussed in the main-text, FashionMNIST [25], Caltech101 Silhouettes, small-NORB dataset [26], a human genome dataset [27] (with similar dimensions as MNIST but more structured), the high-quality CelebA [28] projected in black and white, and in low definition but in color, and CIFAR [29]. The analysis of most of the last datasets are discussed only in the SM and show a similar behavior to the one observed on MNIST.

Our paper is organized as follows. RBMs, their general principles and learning equations are defined in sec. 2. We review previous related works in sec. 3. In sec. 4 we define the different observables used to monitor the training and sampling. Finally, we discuss experimental results in sec. 5.

## 2   Definition of the model

An RBM is a Markov random field with pairwise interactions defined on a bipartite graph of two non-interacting layers of variables: the visible nodes, $\boldsymbol{v} = v_i, i = 1, ..., N_v$, represent the data, while the hidden ones, $\boldsymbol{h} = h_j, j = 1, ..., N_h$, are the latent representations of this data and are there to build arbitrary dependencies between the visible units (see sketch in fig. 1–A). Usually, the nodes are binary-valued in $\{0, 1\}$, yet Gaussian or more arbitrary distributions on real-valued bounded support are also used, ultimately making RBMs adaptable for more heterogeneous datasets. Here, we deal only with binary $\{0, 1\}$ variables for both the visible and hidden nodes. Other approaches

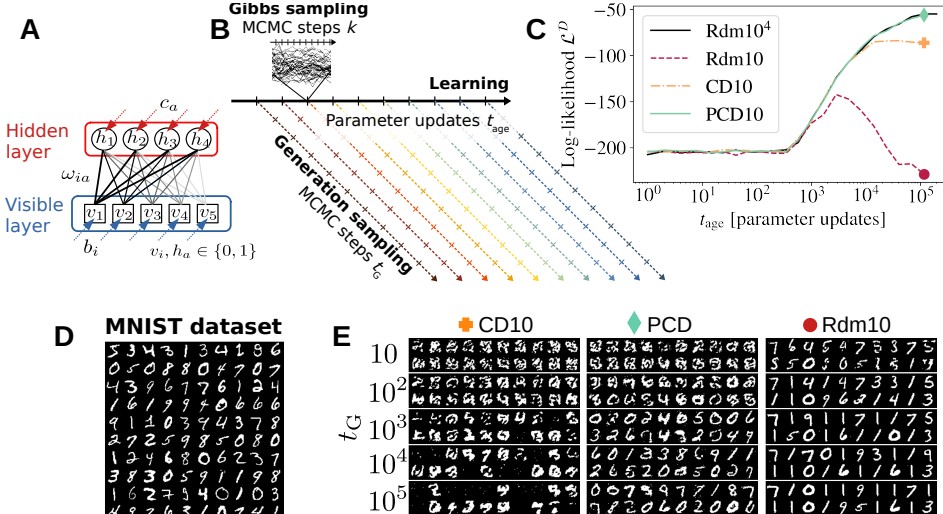

Figure 1: Sketch of an RBM in **A** and of the pipeline of our analysis in **B**. **C** Evolution of the LL of the dataset during the learning for 4 different RBMs trained using either a very long MCMC sampling to estimate the gradient ($k = 10^4$, Rdm-$10^4$) or short samplings $k = 10$ following different recipes for the initialization of the MC chains (CD-10, PCD-10 and Rdm-10). The large symbols mark the LL of the RBMs used to generate new samples in **E**. In **D** we show a subset of the MNIST database, and in **E** a subset of the samples drawn by the $k = 10$ RBMs marked in **C** through a MCMC sampling of the equilibrium Gibbs measure initialized at random and after $10, 10^2, 10^3, 10^4$ and $10^5$ MCMC steps. Note that in our setup the MC chain are initialized at random, and we can observe that RBMs trained with CD-10 do not converge to any digit despite reaching higher value of the LL than Rdm10 which instead manage to create digits quite fast. Also note the lack of diversity of Rdm10 for large sampling time.

using truncated-Gaussian hidden units [30], giving a ReLu type of activation functions for the hidden layer do work well, but in our experiments we observed qualitatively the similar dynamical behavior and therefore we will stick to binary hidden units for the rest of the article. The energy function of an RBM is taken as

$$E[\boldsymbol{v}, \boldsymbol{h}; \boldsymbol{w}, \boldsymbol{b}, \boldsymbol{c}] = -\sum_{ia} v_i w_{ia} h_a - \sum_i b_i v_i - \sum_a c_a h_a, \tag{1}$$

with $\boldsymbol{w}$ the weight matrix and $\boldsymbol{b}$, $\boldsymbol{c}$ the visible and hidden biases, respectively. The Boltzmann distribution is then given by

$$p[\boldsymbol{v}, \boldsymbol{h} | \boldsymbol{w}, \boldsymbol{b}, \boldsymbol{c}] = \frac{\exp(-E[\boldsymbol{v}, \boldsymbol{h}; \boldsymbol{w}, \boldsymbol{b}, \boldsymbol{c}])}{Z} \quad \text{with } Z = \sum_{\{\boldsymbol{v}, \boldsymbol{h}\}} e^{-E[\boldsymbol{v}, \boldsymbol{h}]}, \tag{2}$$

being the partition function of the system. RBMs are usually trained using gradient ascent of the LL function of the training dataset $\mathcal{D} = \{\boldsymbol{v}^{(1)}, \cdots, \boldsymbol{v}^{(M)}\}$, being the LL defined as

$$\mathcal{L}(\boldsymbol{w}, \boldsymbol{b}, \boldsymbol{c} | \mathcal{D}) = M^{-1} \sum_{m=1}^M \ln p(\boldsymbol{v} = \boldsymbol{v}^{(m)} | \boldsymbol{w}, \boldsymbol{b}, \boldsymbol{c}) = M^{-1} \sum_{m=1}^M \ln \sum_{\{\boldsymbol{h}\}} e^{-E[\boldsymbol{v}^{(m)}, \boldsymbol{h}; \boldsymbol{w}, \boldsymbol{b}, \boldsymbol{c}]} - \ln Z. \tag{3}$$

The gradient is then composed of two terms: the first accounting for the interaction between the RBM's response and the training set, and same for the second, but using the samples drawn by the machine itself. The expression of the LL gradient w.r.t. all the parameters is given by

$$\frac{\partial \mathcal{L}}{\partial w_{ia}} = \langle v_i h_a \rangle_{\mathcal{D}} - \langle v_i h_a \rangle_{\mathcal{H}}, \quad \frac{\partial \mathcal{L}}{\partial b_i} = \langle v_i \rangle_{\mathcal{D}} - \langle v_i \rangle_{\mathcal{H}} \quad \text{and} \quad \frac{\partial \mathcal{L}}{\partial c_a} = \langle h_a \rangle_{\mathcal{D}} - \langle h_a \rangle_{\mathcal{H}}, \tag{4}$$

where $\langle f(\boldsymbol{v}, \boldsymbol{h}) \rangle_{\mathcal{D}} = M^{-1} \sum_m \sum_{\{\boldsymbol{h}\}} f(\boldsymbol{v}^{(m)}, \boldsymbol{h}) p(\boldsymbol{h} | \boldsymbol{v}^{(m)})$ denotes an average over the dataset, and $\langle f(\boldsymbol{v}, \boldsymbol{h}) \rangle_{\mathcal{H}}$, the average over the Boltzmann measure in Eq. 2.

A large part of the literature on RBMs focuses on schemes to approximate the r.h.s. of Eqs. 4. In theory, these terms can be computed up to an arbitrary precision using parallel MCMC simulations as long as the number of MC steps are large enough to ensure a proper sampling of the equilibrium configuration space. In other words, sampling times should be larger than the MCMC MT. A naive way to implement this idea is to initialize each of these parallel Markov chains on random conditions, perform $k$ MCMC steps, and use the final configurations to estimate the gradient. We call this scheme **Rdm-k**. This scheme is not used in practice because it would require too many $k$ steps if the MT is too large. Many recipes have been proposed to shorten this sampling and approximate the negative term of the gradient, but since the introduction of the so-called contrastive divergence (CD) by Hinton [31], only a limited number of schemes are used in practice. The first one, **CD-k**, initializes the MCMC simulation from the same data of the mini-batch used to compute the gradient, and performs $k \sim \mathcal{O}(1)$ sampling steps. This approximation relies on the idea that the dataset must be a good approximation of the equilibrium samples of a well trained RBM, something that is not true for a poorly trained machine, whose typical samples are quite distant from the dataset. In the second commonly used method, the last configurations of the Markov chains are saved from one parameter update to the other, and then used to initialize the subsequent chain. As in the other methods, $k$ MCMC steps are used to estimate the gradient for each update. This method is known as persistent-CD: **PCD-k** [32], and takes advantage of the fact that the RBM's parameters change smoothly during the learning and the same is expected for the models' equilibrium distribution[1]. Finally, other approximations, such as mean-field or TAP equations [10], or more elaborate approaches, such as the parallel tempering technique [33, 34], can be used. A common feature of all these recipes is that they rely on very few simulation steps to sample the Boltzmann distribution, and the trained model often generates rather bad configurations.

To illustrate this last comment, let us discuss a provocative experiment shown on fig. 1–E where we try to sample new digits just by sampling the equilibrium measure of our trained RBMs using a MCMC initialized uniformly at random. Let us stress that this is not the standard set-up used in the field to generate new samples, but it is a strong test of the generation power of the trained model. We compare the outcome of the generated samples (as function of the sampling time) for three RBMs trained each following different recipes to compute the negative term of the gradient: CD-10, PCD-10 and Rdm-10. At this point, $k = 10$ might seem a very small number. It is however the typical order of magnitude used in the Literature. The details of the learning protocol are discussed later in the text and summarized in the SM. The first striking point on fig. 1–E, is the inability of the CD-10 RBM to properly sample digits in this set-up. This problem has been reported before [35, 8], but since CD is part of the standard methods, we want to insist on this point. In addition, PCD-10 trained RBM generates proper digits but only after a very long sampling time, while with Rdm-10, quite surprisingly, digits are properly generated after only 10 MCMC steps. Yet, samples obtained after much longer MCMC steps have unbalanced digits representations, like the 1 being over-represented. In fact, we observe that poorly trained models generate samples that do not respect the correct balance between the 10 digits present in the original database (see fig. 1–D). We already mentioned the 1s in the Rdm-10 run, but it happens the same (to a lesser extent) with the 0 in the PCD-10 case. Alternatively to these generated samples, we show the evolution of the LL during the learning of these same three RBMs in fig. 1–C. By definition of the loss function, the LL is intended to be maximized, so it constitutes a traditional measure of the quality of the training. The LL of the models used for the sampling of fig. 1–E are highlighted in large symbols. This plot illustrates the lack of reliability of the LL to monitor the quality of the learning. For instance, CD-10 reaches quite high values of LL and yet, it cannot generate a single digit. For the Rdm-10 is the other way around, it generates decent samples but with a very poor LL. Finally, we observe that the LL of PCD-10 reaches very high values, comparable to RBMs trained using up to $k = 10^4$ MCMC steps to estimate the gradient. We will come back to this point. Our experiment highlights, first, that the LL cannot be used all alone to quantify the generative power of an RBM, and second, that the quality of the generated samples may variate in a non-monotonic way during the generation sampling.

---

[1]This adiabatic scheme is thus valid as long as the relaxation time for the RBM parameters during the learning is much larger than the MCMC mixing times.

# 3 Related works

Recent years have witnessed a large number of studies trying to understand and control the learning of RBMs. In these works, meta-parameters are varied at the same time that an approximation of the LL [8, 36] of the train/test set is monitored throughout the learning process. The stability of the evolution of the LL and the maximum value achieved, are then used to compare the different recipes and parameters. For instance, in Ref. [8] the LL of different learning schemes are evaluated and the samples inspected visually. Unfortunately, no estimator is used to assess the quality of the samples, neither is clear the precise protocol used to generate these samples. In Ref. [9], the authors investigate the stability of the learning of RBMs with binary inputs[2] by comparing the learned features. They show (using mostly the LL and an eye inspection of the learned features) that a centered version of the gradient can be much stabler. In Ref. [7], the authors study, in a systematic way, the convergence properties of CD, PCD and PT on several small toy models that can be analyzed exactly, that is, where the LL can be computed by brute-force. We can find more recent works [37, 38, 39, 40] improving the learning scheme for RBMs and yet, still not giving much information about the quality of the generated samples nor the equilibrium properties of the trained models. In our results below, we will show that without putting on the table this information, the comparison between methods or tuning of parameters, becomes extremely unstable. Finally, we want to mention that it is particularly surprising that the CD recipe (with short $k$) is still used [41, 42, 43, 39]. As shown, RBMs trained with CD are not able to generate proper samples from scratch. And this behavior is true even if doing $k \approx O(100)$ MC steps at each update. The classical explanation for this is that CD is creating local energy well around the datapoint which are separated by large energy barriers [35], since the dynamics do not have time to relax when $k$ is small.

In the rest of the paper, we explain how, by characterizing the MT, we are able to explain the capricious behavior of RBMs with respect to the different usual schemes. Furthermore, we will see that our results are totally consistent to what is observed in other families of EBMs such as ConvNet generative networks [23, 22] or Boltzmann Machines [44, 45] when trained using short $k$ negative chains, which suggests that this explanation applies to EBMs in general.

# 4 Monitoring the learning performance of RBMs

In this work we wish to address the following questions: (i) How can one quantify that an RBM has correctly learned a dataset? (ii) has it learned the equilibrium model? (iii) when should one stop the learning and the sampling of new configurations? We already argued in sec. 2 that, alone, the LL of the dataset is not precise enough to address these questions. For this reason, we have considered several additional observables to quantify the generated samples' quality. We introduce shortly below the different metrics used in our analysis, the precise definitions of all of them can be found in the SM. All the observables are computed and averaged using sets of $N_s = 1000$ true or generated samples.

**Log-likelihood $\mathcal{L}$:** On the contrary to other machine learning methods (such as classification), the LL cannot be exactly evaluated in an RBM without an exhaustive search of the configuration space. Thus we estimate the partition function using the Annealed Importance Sampling (AIS) method [46]. Details of our implementation can be found in SM. We call $\mathcal{L}^D$ the LL of the (original) dataset with respect to the model, and $\mathcal{L}^{RBM}$, the LL of the generated samples with respect to the model.

**Error of the second moment $\mathcal{E}^{(2)}$:** we compute the mean square error (MSE) between the covariance matrix computed with a set of samples generated by the RBM and one of the dataset.

**Error in the power spectral density $\mathcal{E}^{PSD}$:** For images, we compute the log average power spectrum function of radial wave number distance for both a subset of the dataset and the generated set, and then compute the MSE between the two functions. In practice, we compute the error (between true and generated samples) on the logarithm of the squared module of the image's discrete 2D-Fourier transform elements $A_{kl}$ at a fixed distance $d$ in Fourier space

$$P(d) = \log\left(\langle \|A_{kl}\|^2 \rangle_{(k,l)|k^2+l^2=d^2}\right). \tag{5}$$

---

[2]It has been observed in MNIST that by flipping the input $0 \longleftrightarrow 1$, the usual gradient ascent was much slower in one case than in the other.

**Error in the Adversarial Accuracy Indicator (AAI) $\mathcal{E}^{\mathbf{AAI}}$:** The AAI was recently introduced in Ref. [47] with the goal of quantifying the resemblance and the "privacy" of data drawn by a generative model with respect to the training set. This metric is based on the idea of data nearest neighbors. We begin by defining two sets containing, each one, $N_s$ samples: (i) a "target" set containing only generated samples $T \equiv \{\boldsymbol{v}_{RBM}^{(m)}\}_{m=1}^{N_s}$ and (ii) a "source" with samples from the dataset $S \equiv \{\boldsymbol{v}_{\mathcal{D}}^{(m)}\}_{m=1}^{N_s}$. Then, for each sample $m$ in groups $\{T, S\}$, we compute the closest sample —using the euclidean distance– to the set groups $\{T, S\}$. For instance, the closest samples in $\{S\}$ for a sample $m$ in $\{T\}$ would be $d_{TS}(m) = \min_n \left\| \mathrm{v}_{\mathrm{RBM}}^{(m)} - \mathrm{v}_{\mathcal{D}}^{(n)} \right\|$. We can compute in the same way $d_{ST}(m)$, $d_{SS}(m)$ and $d_{TT}(m)$. Once we have all these 4 distances, we can estimate how often the "nearest neighbor" (the sample at the shortest distance) of each sample is contained in the same starting set, or in the other one. When the generated samples are statistically indistinguishable from real ones, both frequencies should converge to the value of a random guess $0.5$ when $N_s$ large. We therefore compute the MSE of these two frequencies with respect to the asymptotic value $0.5$, which defines $\mathcal{E}^{\mathrm{AAI}}$. See SM for more details.

**Relative entropy $\boldsymbol{\Delta S}$:** We approximate the entropy of a given set of data by its byte size when compressed using gzip [48]. In particular, we compute the entropy of a group of $N_s$ random samples of the dataset, $S^{\mathcal{D}}$, and the entropy of another set of data of the same size, called $S^{\mathrm{cross}}$, composed in equal portions by samples of the train and generated set. Finally, we compute the relative size of this cross set with respect to the dataset's one, minus the expected value for identical sets, i.e. $\Delta S = S^{\mathrm{cross}}/S^{\mathcal{D}} - 1$. Then, a large $\Delta S$ indicates us that the generated set is less "ordered" than the dataset. On the contrary, a small $\Delta S$ reflects that the generated samples lack diversity.

In the SM, we show the evolution of other quality images observables along the generation sampling (such as the error in the third moment correlations, the error in the energy, or the Frechet Inception Distance score [49] for images). All them showing very similar trends to the ones shown here.

## 5 Results

Equipped with this set of quality measures, we now want to monitor and evaluate the learning process. To this end, we need to control precisely the learning dynamics and its interplay with the equilibration time. To do so, we try to isolate all the different factors affecting the learning, and for this reason, at a first stage, we avoid using the common tricks to speed up the estimation of the r.h.s. of the gradient of Eqs.4 during the learning. With this goal, we focus our first experiments on the Rdm-k schemes, and discuss the interplay with standard methods such as CD or PCD later on. The Rdm-k scheme is very practical to study the convergence to equilibrium for two reasons: (i) the MCMC chains are initialized from configurations that are uncorrelated with the dataset, and (ii) the sampling protocol used during the learning is identical to the one used to generate new samples from scratch with the learned RBM. The pipe-line of our experiments is summarized in Fig. 1–B. We split the analysis in two separated stages: learning and generation sampling. During the learning, we update the parameters for $\sim 2 \cdot 10^5$ times using the gradient estimated following the Rdm-k scheme (with $k = 10, 50, 100, 500, 10^3$ and $10^4$). We save the RBM parameters at different numbers of updates equally spaced in logarithmic scale. The number of updates used to train a particular RBM will be referred as its *age*, $t_{\mathrm{age}}$. During the generation stage, we sample the equilibrium Gibbs measure of each RBM (of age $t_{\mathrm{age}}$) with a MCMC initialized at random. We evaluate the quality of the samples generated at Gibbs sampling times $t_{\mathrm{G}}$ equally spaced in logarithmic scale. Unless something else is mentioned, we will consider RBMs with $N_{\mathrm{h}} = 500$ hidden nodes and trained with a fixed learning rate of $\eta = 0.01$, hence the different RBMs will differ only by the value of $k$ and $t_{\mathrm{age}}$. Further details on the learning parameters can be found in SM.

**Evolution of the quality of the generated samples with the sampling time—**

We show in Fig. 2–A the evolution of the quality estimators of the generated samples throughout the generation sampling for RBMs of different ages (i.e. following the different color lines in Fig. 1–B), and trained with three different $k$. We recall that perfect samples, in the sense that they are indistinguishable from the dataset, would have a score 0 in all the quality observables except for the LL, where $\mathcal{L}^{\mathrm{RBM}}$ should match $\mathcal{L}^{\mathcal{D}}$ ($\mathcal{L}^{\mathcal{D}}$ is marked as a horizontal line). For all $k$ and early learning times (ages below $\lesssim 10^3$ updates in red-yellowish colors), all the metrics describe a constant behavior with $t_{\mathrm{G}}$ at roughly the same values obtained using a completely untrained RBM, and generally far away

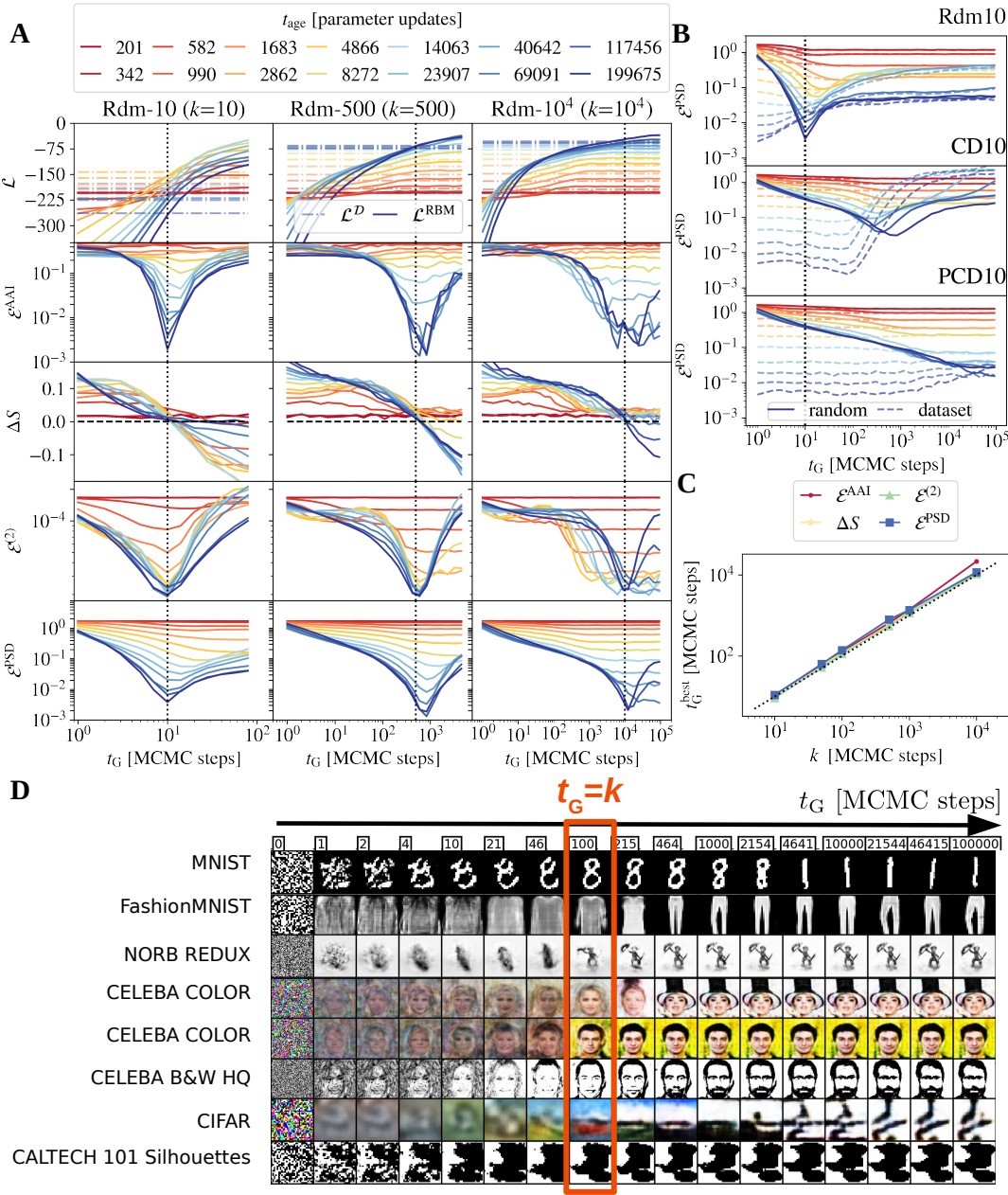

Figure 2: Evolution of the quality of the generated samples during the data generation sampling (along the color dashed lines of Fig. 1–B). In **A** we show the LL and the rest of quality estimators (introduced in Sect. 2) as function of the sampling time, $t_G$, for chains initialized at random and using RBMs trained for an increasing number of updates $t_{age}$ (age is coded by colors) using the Rdm-k scheme with $k = 10, 500, 10^4$. In general, the closer $\mathcal{E}$ and $\Delta S$ are to zero, the more similar the generated and original datasets are. For the LL, we show the value computed with the generated set ($\mathcal{L}^{RBM}$, in continuous growing lines) and with the original dataset ($\mathcal{L}^{\mathcal{D}}$, in dashed horizontal lines). We observe that these two measures cross at roughly the same $t_G$ at which all $\mathcal{E}$ reach a minimum and $\Delta S$ crosses zero. **B** We show the evolution of $\mathcal{E}^{PSD}$ for 3 RBMs trained using the Rdm-10, CD-10 and PCD-10 recipes, for the same ages as in **A**. We consider two possible initializations of the Markov chain (random in full lines and dataset in dashed lines). **C** We plot the position of the best performance peak and the time at which $\Delta S = 0$, as function of the number of MCMC steps $k$ used for estimating the negative term of the gradient during the training. **D** We show an example of the images generated as function of $t_G$, using RBMs trained with Rdm-100 and with different datasets. The sampling time $t_G$ that matches $k$ is hightlighted in an orange frame.

from 0. For an intermediate range of ages (pale blue colors), we observe an important improvement of the metrics with the RBM's age, up to a certain value (dark blue lines) above which time evolution curves start to collapse. Furthermore, for $k = 10$, only young ages converge to constant curves in $t_G$, something that extends to intermediate ages for $k = 500$ and covers the majority of the ages for $k = 10^4$. In all the cases, the sampling of the oldest machine ($t_{age} \sim 200k$ updates) describe a non-monotonous behavior in the quality measures, with a clear peak of best quality at $t_G \sim k$. This is also the point where $\mathcal{L}^D$ and $\mathcal{L}^{RBM}$ intersect and where $\Delta S$ crosses 0. In Fig. 2–C, for each learning process using a different $k$, we plot the position of this $t_G^{best}$ (obtained for each quality metrics) against the value of $k$ used to train the machine. We observe a perfect match between both times. At this point we want to stress that we obtain exactly the same qualitative behavior with the other considered datasets, as shown in the SM. Fig. 2–C reflects that these RBMs "memorize" the number of MCMC steps used to estimate the gradient during the learning phase. Memory effects are a clear signal of non-equilibrium effects [50] and in fact, this effect is not present as long as $k$ is long enough to equilibrate the system during the learning. Indeed, we see that the larger $k$, the longer (in terms of number of updates) it takes to observe memory effects (for $k = 10^4$ we only appreciate out-of-equilibrium effects above 100k updates, while they appear around $t_{age} \gtrsim 10^3$ updates in the case of $k = 10$). These two regimes (equilibrium and OOE) are also reproduced when using the CD and PCD schemes during the learning. We show in Fig. 2–B the evolution of $\mathcal{E}^{PSD}$ of the generated samples as function of $t_G$ for 3 RBMs trained with the 3 schemes and $k = 10$ steps (higher values of $k$ are shown in the SM). We compare results with MCMC chains initialized either at random (continuous lines) or at the dataset (dashed lines). The sampling of the RBMs trained with CD-10 is also highly complex in time, and much slower than that of the Rdm-10 considering that the values of $\mathcal{E}^{PSD}$ measured in runs from random initialization and from the dataset never converge to the same values for large $t_{age}$. The situation is rather different for PCD-10, where the error sampling curves describe a very smooth behavior in time and a very slow convergence towards an equilibrium value. Yet According to the minimum values reached in $\mathcal{E}^{PSD}$, the equilibrium samples of that model are significantly worse than the best samples created using the Rdm scheme (even working in the OOE regime). This difference is observed also in the rest of quality estimators and at much higher values of $k$, as shown in Fig. 5 of the SM. Indeed, for MNIST, one needs to reach PCD-$10^3$ to start to match the qualities achieved with the Rdm-10 recipe when exploiting the OOE regime.

**Quantifying the mixing time:** At this point, we try to quantify the MT or relaxation time of the MCMC dynamics of our RBM as a function of $t_{age}$, to rationalize figs. 2. To do so, we compute the time-autocorrelation function, $\rho$, of the averaged visible variables

$$\rho(t) = \frac{C_m(t)}{C_m(0)}, \text{ where } C_m(t) = \frac{1}{N_v} \sum_i (m_i(t) - \bar{m}_i)(m_i(0) - \bar{m}_i) \qquad (6)$$

where $m_i(t) = \text{sigmoid}(\sum_a w_{ia} h_a(t) + b_i)$ and $\bar{m}_i$ is the equilibrium magnetization.[3] At this point, let us stress that in a perfectly trained RBM, $\bar{m}_i$ would be equal to $1/N_v \sum_m^M v_i^{(m)}$. Yet, this is not the case in a poorly trained machine. For this reason, we estimate these equilibrium values from a long $10^5$ MCMC steps simulation. The value of $\rho(t)$ will be close to 1 if the data at $t = 0$ and $t$ are very similar, and to 0 if they are completely decorrelated. In order to measure properly the MT, we need to compute the autocorrelation function from equilibrium configurations. Therefore, we measure $\rho$ with Eq. (6) but after discarding the first MCMC $\approx 10^4$ steps[4]. We show the time-correlation curves obtained for our best trained machine (using Rdm and $k = 10^4$) in Fig. 3–A for different values of $t_{age}$. Clearly, the older the machine, the slower the relaxation. Such a study is harder with the poorly trained RBMs because equilibration is achieved at times $t_G > 10^5$ for most of the $t_{ages}$, being particularly dramatic for Rdm-10 where relaxations are extremely slow, see SM. The decay being typically exponential, we can easily extract the MT (or the exponential relaxation time), $t_\alpha$, from a fit of $\rho(t)$ to $A \exp(-t/t_\alpha)$. We show the dependence of $t_\alpha$ with $t_{age}$ in Fig. 3–C. This $t_\alpha$ (for the slowest observable) controls the time-scale of the relaxations in the system, the time necessary to sample independent (uncorrelated) configurations, which means that the number of MCMC iterations necessary to reach equilibrium from the beginning of a MCMC simulation is expected to scale with it. Typically, the thermalization time, $t_{therm}$, is some few times this $t_\alpha$ [51]. This means that, in order to guarantee a sampling of equilibrium configurations,

---

[3]The time-autocorrelation function of the visible, hidden or averaged hidden units are qualitatively the same.
[4]In practice, we discard as many MCMC steps as needed to stop detecting dependence on the $t = 0$ point.

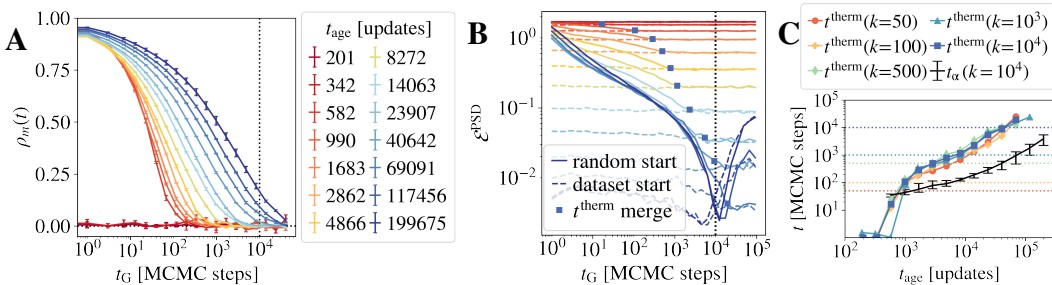

Figure 3: **A** The time-autocorrelation function of the visible mean magnetizations, $\rho_m$ (see Eq. (6)), computed during the sampling of an RBM trained with Rdm-$10^4$ for different number of updates $t_{\mathrm{age}}$. This function carries the information about the number of MCMC steps that are needed (in equilibrium) to forget a initial configuration of the MC chain. Clearly, the time needed to decorrelate grows with the age of the RBM. In **B**, for the same RBMs, we show the evolution of the generated samples' quality through the $\mathcal{E}^{\mathrm{PSD}}$ observable, as function of the sampling time $t_G$, and for two different initializations: from the dataset and from random. We mark the times $t^{\mathrm{therm}}$ beyond which both runs converge to the same constant value. **C** We compare the mixing times, $t_\alpha$, obtained from a fit of the curves $\rho_m(t)$ in **A** to the form $\exp(-t/t_\alpha)$, with the convergence-to-equilibrium times $t^{\mathrm{therm}}$ obtained in **B** for RBMs trained with different values of $k$. We see that the equilibration time $t^{\mathrm{therm}}$ follow the same trend for various values of $k$.

one needs Gibbs samplings longer than this $t_{\mathrm{therm}}$ (which scales with $t_\alpha$). Note that $t_{\mathrm{therm}}$ (the distance to equilibrium) might depend on the MC starting point while $t_\alpha$ should not, yet $t_{\mathrm{therm}}$ will grow proportionally with $t_\alpha$ for any learning scheme. In fact, we can estimate independently the thermalisation time for the Rdm-k runs, $t_{\mathrm{therm}}$, by comparing results from two independent simulations with different initializations: random and dataset, and wait until they converge to the same equilibrium expectation value. This means that one can estimate $t_{\mathrm{therm}}$ (for each $t_{\mathrm{age}}$) by the number of MCMC steps needed to collapse the measures of the two trajectories to a constant value. See for instance Fig. 3–B, where we show the evolution of $\mathcal{E}^{\mathrm{PSD}}$ with $t_G$ for the two runs. The merging point for each $t_{\mathrm{age}}$ is highlighted with a square. We show the $t_{\mathrm{therm}}$ values we obtain with this procedure in Fig. 3–C for the different values of $k$.

As a result, these two times, $t_\alpha$ and $t_{\mathrm{therm}}$, provide us with a lower and a safe upper bound (for instance, when initializing at random) for the minimum number of MCMC steps (for the learning's Gibbs sampling), $k$, needed to learn a good equilibrium RBM model. As shown in Fig. 3–C, these times grow progressively with $t_{\mathrm{age}}$, and in particular, falling always out of the range of iterations used to train the $k = 10$ runs (once the RBM starts to learn the dataset for $t_{\mathrm{age}} \gtrsim 10^3$ updates), and falling out the range of scheme Rdm-500 for $t_{\mathrm{age}} \gtrsim 10^4$ and for the last 2 $t_{\mathrm{age}}$ for the Rdm-$10^4$ scheme (i.e. for $t_{\mathrm{age}} \gtrsim 10^5$ updates). In other words, at those ages, the learning enters in the OOE regime and the RBM starts to learn the dynamical process rather than the equilibrium properties. In fact, these ages mark the crossover between smooth and non-monotonic evolution of the observables in Fig. 2–A. This reinforces our previous claim that the apparition of the "best performance peak" in the error estimators is a purely non-equilibrium effect. A similar comment can be made for the LL. See for instance Fig. 2–A first row, for each $t_{\mathrm{age}}$, we compare the LL measured on the generated $\mathcal{L}^{\mathrm{RBM}}$ and original dataset $\mathcal{L}^{\mathcal{D}}$ as function of its generation sampling time $t_G$. We distinguish, again, two distinct behaviours associated to the equilibrium or OOE regime. In the first case, the LL of the generated samples converges to the value of the dataset and remain there even at large $t_G$. Whereas, in the OOE regime, $\mathcal{L}^{\mathrm{RBM}}$ crosses $\mathcal{L}^{\mathcal{D}}$ at $t_G \approx k$ and continues growing afterwards. This highlights the problem of using the LL to evaluate the sample generation power. While the fact that the likelihood increases during the learning is in general a good sign, it is not necessarily linked with the equilibrium properties of the model. In fact, the observation that $\mathcal{L}^{\mathrm{RBM}}$ tends to reach much higher values than $\mathcal{L}^{\mathcal{D}}$ should be rather interpreted as a sign of poor learning, as shown in Fig. 1.

As a direct consequence of the results discussed above, we can see that even in the case where sampling is openly OOE, such as with the Rdm-10 scheme, one can generate fairly good samples as long as one limits the generation sampling to $t_G \sim k$, as shown in Fig. 1 for $k = 10$: digits are reproduced with uniform probability, and they look OK on visual inspection. For longer samplings

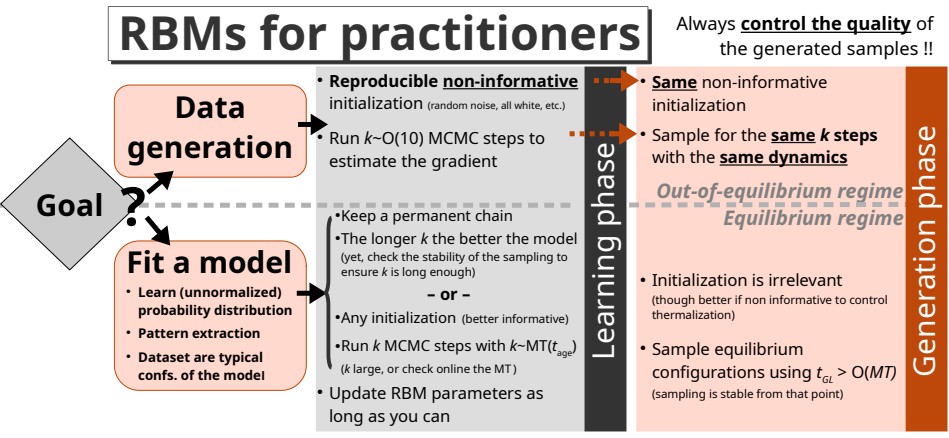

Figure 4: Flow chart of the algorithm proposed to train and sample properly an RBM depending on the final goal of the training.

we start to get clear unbalanced distributions. It is also quite remarkable that when matching $t_{\mathrm{G}}$ with $k$ in the OOE regime of Rdm schemes, we get better quality samples than with the standard CD o PCD schemes even at very long sampling times, as we can see from the comparison of $\mathcal{E}^{\mathrm{PSD}}$ in Fig. 2–B. This is also true for other quality observables or larger values of $k$ as shown around Fig. 5 of the SM. In particular, for NMIST, the OOE recipe always generates better samples than the equilibrium regime reached with PCD trained with $k < 10^3$ steps.

## 6 Conclusions

In this work, we show that the training of an RBM is richer and more complex than what previously thought. We show that the choice of the number of MCMC steps, $k$, used to estimate the gradient during the learning has a dramatic impact on the learned model's dynamics and equilibrium states. In practice, we find two regimes: (i) the OOE, when $k$ is much lower than the MCMC mixing time, the trained RBM model encodes the dynamical process followed to train it, resulting in a highly non-linear, unstable and slow evolution of the quality of the generated samples with the sampling time, and a convergence towards poor and strongly biased samples. And (ii) the equilibrium regime, when $k$ overpasses largely the mixing time, the marginal equilibrium probability distribution of the trained model (over the visible layer) matches that of the dataset. In such case, the quality of the generated samples improves smoothly in time converging faster to the equilibrium values, converging much faster than in the OOE regime. While regime (ii) is needed to obtain good trained models, we show that regime (i) is particularly well suited for sample generation because the data sampling non-linearities can be controlled in the Rdm-k approach, and exploited to generate high-quality data in short training times. We further show that the MC mixing time increases significantly as the learning progresses, being always much larger than the standard $k$s used in the Literature. We summarize the best strategy to train and exploit RBMs depending on their purpose in Fig. 4.

The existence of a crucial competing mechanism in the training process had already been noted and exploited in EBMs with more complex energy functions (those defined with deep convolutional neural networks) [22, 23], and after our results, both regimes were also found in the simpler brother of the RBMs, the Bolzmann Machines [44, 45]. This evidences that the OOE regime, which was widely unnoticed in the Literature for decades, is a general consequence of the MCMC based likelihood maximization methods, so it is expected to be reproduced in many other popular models like Deep Boltzmann Machine (DBM). Interestingly, our results should simplify their training and use, at least in the OOE regime. More generally, this work provides a set of key points to be checked when working with EBMs, and a new fundamentally distinctive attribute to theses generative models: equilibrium or OOE. We leave for future works systematic comparisons between the models trained in each of these regimes, and in particular, the effect on the features. For instance, we observed that RBMs trained with low $k$ suffer strong aging effects extremely similar to those observed on the spin glasses [52]. Effects which are not present in equilibrium RBMs.

## Acknowledgments and Disclosure of Funding

A.D. and B.S. were supported by the Comunidad de Madrid and the Complutense University of Madrid (Spain) through the Atracción de Talento program (Refs. 2019-T1/TIC-13298 and Ref. 2019-T1/TIC-12776). BS was also partially supported by MINECO (Spain) through Grant No. PGC2018-094684-B-C21 (partially funded by FEDER).

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
