# Supplementary material

**Aurélien Decelle**[1,2,†]  **Cyril Furtlehner**[2]

**Beatriz Seoane**[1,⋆]

[1]Departamento de Física Téorica, Universidad Complutense, 28040 Madrid, Spain.
[2]Université Paris-Saclay, CNRS, INRIA Tau team, LISN, 91190, Gif-sur-Yvette, France.
[†]`adecelle@ucm.es`; [⋆]`bseoane@ucm.es`

## Contents

35th Conference on Neural Information Processing Systems (NeurIPS 2021).

The code used for the experiments can be found at Github.

# 1  Quality Observables: detailed definitions

We detail here the different observables used to quantify the quality of the generated samples.

## 1.1  Log-likelihood (LL)

The likelihood is the most difficult observable to compute. In practice, the Annealed Importance Sampling (AIS) method [1] is often used, since, as soon as $N_v \gtrsim 30$, an exact enumeration of all the RBM's states becomes impossible. Some recent works have dedicated large efforts to understand this object [2], from which we can extract a generic method to compute a reasonable estimate. First, let us recall that AIS is based on a simulated annealing process where a configuration is gradually brought from temperature $T = \infty$ to $T = 1$ using a set of bridging distributions. We therefore define a temperature schedule: $\{\beta_k \equiv 1/T_k\}$ such that $0 = \beta_0 < \beta_1 < ... < \beta_K = 1$. For each temperature, we define the transition operator, $T_k(\boldsymbol{v'}, \boldsymbol{v})$ to bring a configuration $\boldsymbol{v}$ to $\boldsymbol{v'}$ varying the temperature according to the temperature schedule. In our case it is done using MC sampling layer-wise. In our experiment, we use as bridging distributions the following

$$p_k(\boldsymbol{v}, \boldsymbol{h}) = \frac{1}{Z_k} \exp(-\beta_k E(\boldsymbol{v}, \boldsymbol{h})). \tag{1}$$

For the rest, we follow [2], using a set of $N_I$ initial configurations, and $N_\beta$ equally-spaced intermediate temperatures to compute the estimation of the partition function.

In our work, we used a set of $N_\beta \in [10^4, 10^5]$ temperatures uniformly distributed in this interval (depending on the system size). Then, results are averaged over $N_I \in [1000, 10000]$ realizations.

## 1.2  Energy

Given a trained model, we can compare the mean energy

$$E[\boldsymbol{s}, \boldsymbol{\tau}; \boldsymbol{w}, \boldsymbol{b}, \boldsymbol{c}] = -\sum_{ia} v_i w_{ia} h_a - \sum_i b_i v_i - \sum_a c_a h_a, \tag{2}$$

of the generated and original samples, namely, $E_{\text{RBM}}$ and $E_{\mathcal{D}}$. In a perfect set-up, $E_{\text{RBM}}$ should be larger to $E_{\mathcal{D}}$ at the begining of the generation sampling (when initialising at random) and converge to $E_{\mathcal{D}}$ at long times after equilibration. In practice, one observes that $E_{\text{RBM}}$ goes below $E_{\mathcal{D}}$ at long sampling times if the machine was trained out of equilibrium. In order to quantify the generated sample quality, we find useful to track the error in the energy, $\mathcal{E}^{\text{E}}$, as the mean squared error (MSE) between both measures.

## 1.3  Error of the second moment

Firstly, we compute the averaged covariance matrix of the data representation sites $C_{ij} = \overline{\text{Cov}(v_i, v_j)}$ (with $\overline{(\cdots)}$ the average over $N_s$ independent samples). This matrix can be computed either using the original dataset, in which case we call it $C_{ij}^{\mathcal{D}}$, or the generated dataset, then called $C_{ij}^{\text{RBM}}$. Once having both matrices, we estimate the distance between both kinds of $C_{ij}$ at each

combination $i, j$ through the MSE (which is equivalently the square of the Frobenius norm between the two matrices)

$$\mathcal{E}^{(2)} \equiv \frac{2}{N_{\rm v}(N_{\rm v}-1)} \sum_{i<j} \left(C_{ij}^{\rm RBM} - C_{ij}^{\mathcal{D}}\right)^2 .$$
(3)

## 1.4 Error of the third moment

Similarly, we can compute the MSE of the third moment of the data representation sites $C_{ijk} = \overline{(v_i - m_i)(v_j - m_j)(v_k - m_k)}$, with $m_i = \overline{v_i}$. We will refer to this error as $\mathcal{E}^{(3)}$ and it was not shown in the main-text. Since the number of moments scales as $\sim \mathcal{O}(N_{\rm v}^3)$, we shall focus only on the most active sites: the ones where the empirical frequency $m_i$ is as close as possible to $0.5$. In practice, we took the 50 more active sites. The expression is given by

$$\mathcal{E}^{(3)} = \frac{6}{N_{\rm v}(N_{\rm v}-1)(N_{\rm v}-2)} \sum_{i<j<k} \left(C_{ijk}^{\rm RBM} - C_{ijk}^{\mathcal{D}}\right)^2 .$$
(4)

## 1.5 Power spectrum density

For images, we compute the average power spectrum at fixed distance $r$ of both a set of generated images and of the original dataset. In practice, we compute the logarithm of the module of the Fourier coefficient at a fixed distance $d$ in Fourier space

$$P(d) = \log\left(\langle\|A_{kl}\|^2\rangle_{(k,l)|k^2+l^2=d^2}\right),$$
(5)

where $A_{kl}$ are the image's discrete 2D Fourier transform elements. Again, we compare the values obtained with the generated dataset with those of the real dataset through the MSE. We will refer to this error as $\mathcal{E}^{\rm PSD}$ which is defined as:

$$\mathcal{E}^{\rm PSD} = \sum_d \left[P(d)^{\rm RBM} - P(d)^{\mathcal{D}}\right]^2 .$$
(6)

## 1.6 Adversarial Accuracy Indicator

A recent indicator has been introduced in Ref. [3] for measuring the resemblance and the "privacy" of data drawn from a generative model with respect to the training set that has been used to train the model. This metric is based on the idea of data nearest neighbors. We begin by defining two sets containing each one $N_{\rm s}$ samples: (i) a "target" set containing only generated samples $T \equiv \{s_{RBM}^{(m)}\}_{m=1}^{N_{\rm s}}$ and (ii) a "source" set containing only samples from the dataset $S \equiv \{s_{\mathcal{D}}^{(m)}\}_{m=1}^{N_{\rm s}}$. Then, for each sample $m$ in $T$, we compute the minimum euclidean distance to a sample in $S$, that is $d_{TS}(m) = \min_n \left\|s_{\rm RBM}^{(m)} - s_{\mathcal{D}}^{(n)}\right\|$. In other words, we look for the "nearest neighbor" of our generated sample in the original dataset. We can do exactly the same calculation the other way around, and look for the generated sample that resembles more to a sample in the original set, and compute $d_{ST}(m)$. Finally, we can also measure the minimum distance from a sample $m$ to a different sample in the same set, thus obtaining $d_{SS}(m)$ and $d_{TT}(m)$. Once having all these 4 distances, we can estimate the frequency at which we find the nearest neighbors of a data point in the same set that already contained that point, that is

$$\mathcal{A}_S = \frac{1}{N_{\rm s}} \sum_m^{N_{\rm s}} \mathbf{1}\left(d_{SS}(m) < d_{ST}(m)\right) \quad \text{and} \quad \mathcal{A}_T = \frac{1}{N_{\rm s}} \sum_m^{N_{\rm s}} \mathbf{1}\left(d_{TT}(m) < d_{TS}(m)\right).$$
(7)

It is clear that if the generated samples were statistically indistinguishable form real ones, these two measures should converge to the value of a random guess $0.5$ when $N_s \longrightarrow \infty$. Furthermore, $\mathcal{A}_S$ close to 1 means that the generated samples are very dissimilar to the true ones while, if close to zero, means that the generated samples are that close to the true ones that we are overfitting the dataset. On the other way around, whenever $\mathcal{A}_T$ is close to one, we can affirm that the generated samples are all very close to each others, which can be related to a lack of diversity in the target set or because the generated samples are very far away from the true ones. If close to zero, it reflects again overfitting. In the main text article, for simplicity, we use the MSE estimate of the two indicators:

$$\mathcal{E}^{\rm AA} \equiv (\mathcal{A}_S - 0.5)^2 + (\mathcal{A}_T - 0.5)^2.$$
(8)

## 1.7 Entropy

Our last indicator to compare the true and the generated datasets is the entropy. Here, we will approximate the entropy of a given set of data by its byte size when compressed using `gzip`. We therefore define the entropy of $N_s$ samples from the true dataset as $S_{\mathrm{src}}$ and we compare it to the entropy of a set of the same size where half of the samples are drawn from the true dataset while the other half is taken from the generated configurations. We refer to this entropy as the cross entropy $S_{\mathrm{cross}}$. We compare these 2 quantities through

$$\Delta S = \frac{S_{\mathrm{cross}}}{S_{\mathrm{src}}} - 1 \tag{9}$$

When $\Delta S$ is large, it means that the generated samples are less "ordered" than the samples coming from the source. On the contrary, if small, it means that the generated samples lack diversity (as compared to the original dataset).

## 1.8 Frechet Inception Distance score

For image datasets, we can also use other typical estimators such as the Frechet Inception Distance score (FID) [4]. The FID metric compares the distribution of the activation variables of a neural network (typically Inception v3 trained on the ImageNet) when fed with real or generated images. As we show in Figs. 4 and Fig. 19, this estimator behaves qualitatively in the same way than the estimators discussed in the main-text.

## 2 Details of the RBM used for the training of MNIST

### 2.1 RBMs details

All the RBMs that have been used to learned the MNIST dataset have been trained with the following parameters. First of all, only the first 10000 images of MNIST where used, threshold (for the mapping to black/white) at the value 0.5. The test set consisted of 1000 other images. Parameters of the network and learning procedure:

- Number of hidden nodes: $N_{\mathrm{h}} = 500$.
- Learning rate: $\alpha = 0.01$.
- Minibatch size: $n_{mb} = 500$.
- No $\ell_2$ regularization of momentum.
- The gradient is centered according to Ref.[5].
- The visible biases are initialized to match the empirical frequency of the training dataset:

$$\eta_i = \log\left(\frac{\bar{m}_i}{1 - \bar{m}_i}\right) \text{ where } \bar{m}_i = \frac{1}{M}\sum_m s_i^{(m)}. \tag{10}$$

- The number of Markov chains used to estimate the negative term of the gradient was always equal to $n_{mb}$.
- The number of MCMC steps used for the negative chains is indicated by the variable $k$, and vary in the different experiments.
- For the indicators, we used $N_s = 1000$ samples.

### 2.2 Computer load

All the RBM's trainings were done on NVIDIA GPU RTX 3090, using the pytorch library [6]. The runtime in MNIST ranges from less than an hour when $k = 10$ to $\sim 2$ days when $k = 10^4$, for reaching a number of updates $t_{\mathrm{age}} = 200000$. The generation sampling of new configurations is done on CPU (Intel(R) Xeon(R) @ 2.60GHz). Typical analysis were several minutes long for each $t_{\mathrm{age}}$, which the exception of the time auto-correlation functions that required some few hours to complete.

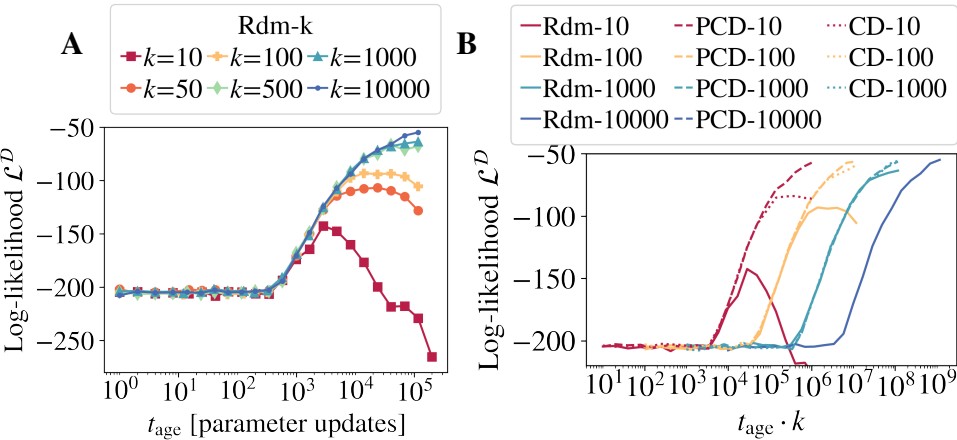

Figure 1: Evolution of the log-likelihood (LL) on the dataset along the learning. In **A** we show the LL measured during the training of RBMs with the Rdm-k scheme, as function of the number of parameters updates, $t_{\text{age}}$. We include data for RBMs trained with different values of $k$. In **B** we compare (for several values of $k$) the LL obtained with different schemes as function of the product $t_{\text{age}} \cdot k$, which is proportional to the total learning time.

## 3   Additional results for MNIST

### 3.1   Rdm-k scheme

We show the evolution of LL on the dataset during the learning process using the Rdm-k scheme for different values of $k$ in fig. 1-A, and compared with other training schemes in fig. 1-B.

We illustrate on the fig. 2 the samples that we obtain at different sampling times, $t_{\text{G}}$, for the older RBM we have trained with the Rdm-100 scheme.

We clearly see that during the learning process, the RBM memorized in its structure the dynamics followed to train it, and that the equilibrium configurations do not match the dataset density distribution. We stress here that the number of MCMC steps used to estimate the negative term of the LL gradient, $k$, is rather large in this case ($k = 100$, for sure larger than the common values used in the literature) and yet, we start to observe out-of-equilibrium effects quite quickly during the learning.

We show on fig. 3 and fig. 4 the evolution of the quality observables measured during the sampling stage for various values of $k$. As described in the main-text: when $k$ is small in comparison to the mixing time the system enters quickly in the OOE regime, developing a peak of performance at $t_G \sim k$. For the case $k = 100$, we show in fig. 5, the evolution of the position of this peak (its position in the time axis in fig. 5-A and its height in fig. 5-B) with the RBM's age. We can see that the position of the best quality peak converges quite early to $t_G \sim k$, but the overall quality of the generated samples at that time continues improving with the number of parameter updates during the training even if the LL of the dataset, $\mathcal{L}^{\mathcal{D}}$, does not improve anymore at these values of $t_{\text{age}}$, as is shown in fig. 1. Furthermore, the quality of the generated samples at this peak does not change much with the value of $k$ used to estimate the negative term of the gradient, as shown in fig. 6-A. And not only, already with some few $k$ steps, we can generate samples that are much better than those generated with CD-k or PCD-k using much larger values of $k$ (see fig. 6-B).

We further compare the evolution of the quality estimator $\mathcal{E}^{\text{PSD}}$ along 2 sampling runs initialized either at random or at the dataset in fig. 7. We highlight the time, $t^{\text{therm}}$, at which both simulations converge to the equilibrium value. These are the times that were plotted as function of $t_{\text{age}}$ in fig.3-C in the main-text.

### 3.2   Relaxations of equilibrium and OOE models

In the main-text, the mixing times were estimated through the equilibrium magnetization time-autocorrelation function $\rho(t)$. At that point, we mentioned that we could only properly estimate

Rdm-100 random start

Rdm-100 dataset start

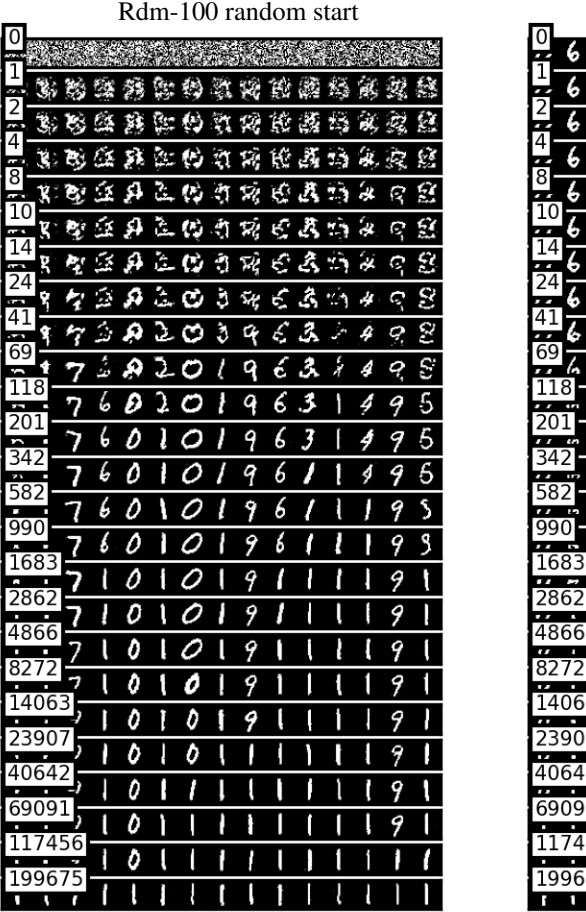
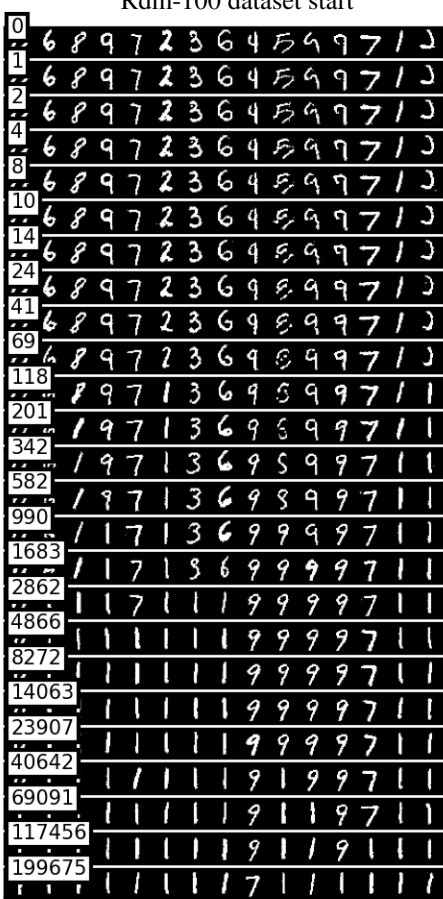

Figure 2: We show some examples of the images we generate at different steps of the data generation sampling of an RBM trained with the Rdm-100 scheme for $t_{\mathrm{age}} = 199876$ parameter updates. Samples generated at the same number of MCMC steps are grouped together in rows and the value of $t_G$ is given in the top left corner of each row. Runs initialized at random configurations are shown on the left panel, and on the dataset on the right. Each column follows the evolution of a Markov chain initialized from random or from the dataset and iterated for almost 200000 steps. We see that at $t_G = 118 \sim k = 100$, we obtain nice samples having a fair representation of all digits. For lower number of MCMC steps, the chain didn't converge yet to numbers (in the case of random initialization) and, for longer ones, almost all samples are vertical ones (for both kinds of initialization) which clearly dominate the learned equilibrium distribution.

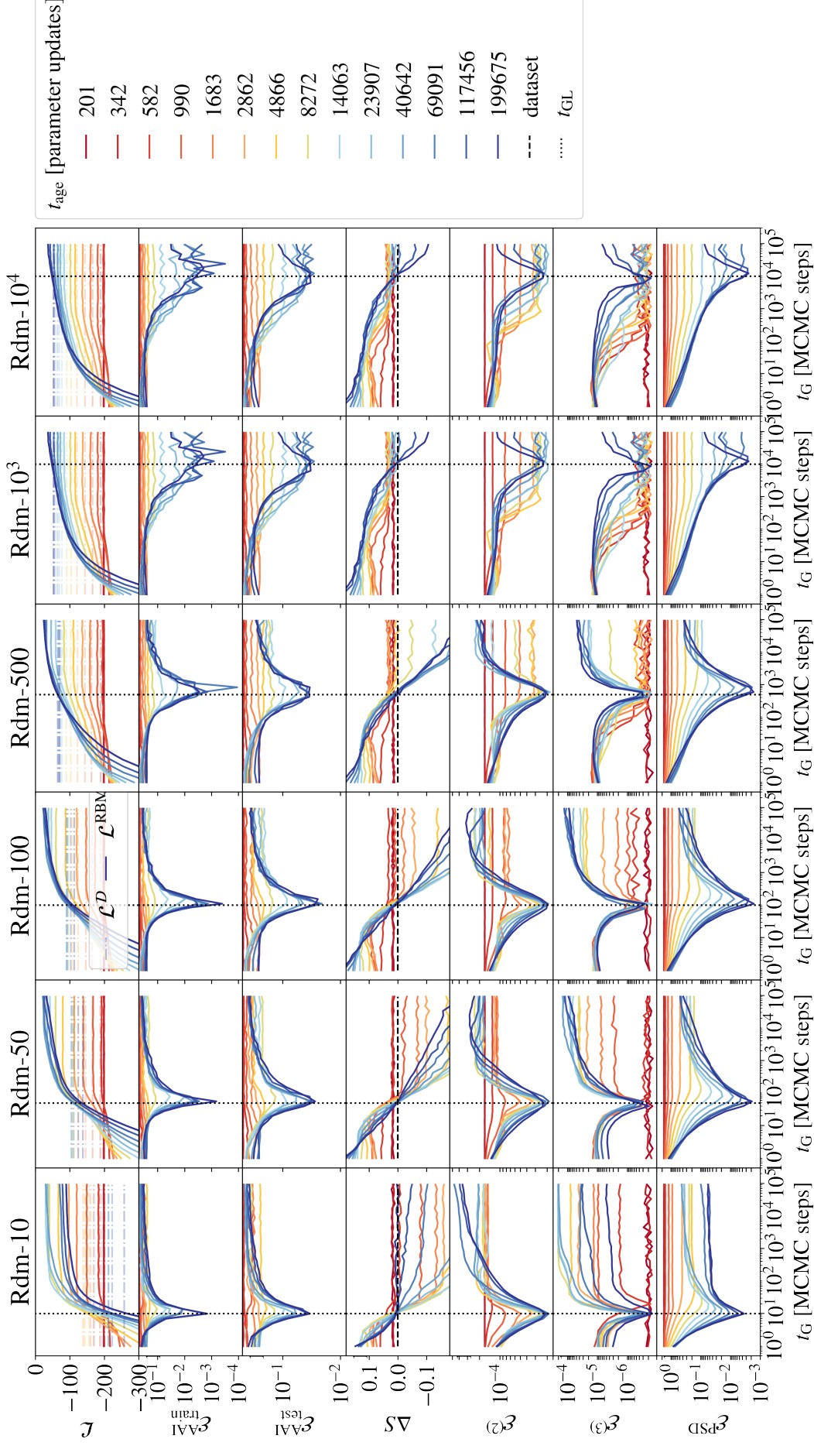

Figure 3: Evolution of the quality estimators along the sample generation for MNIST database for RBMs trained using the Rdm-k approach and with different values of $k$. For the case of the $\mathcal{E}^{\mathrm{AAI}}$ estimator, we have shown both the results obtained when comparing with the training ($\mathcal{E}^{\mathrm{AAI}}_{\mathrm{train}}$) or the test set ($\mathcal{E}^{\mathrm{AAI}}_{\mathrm{test}}$).

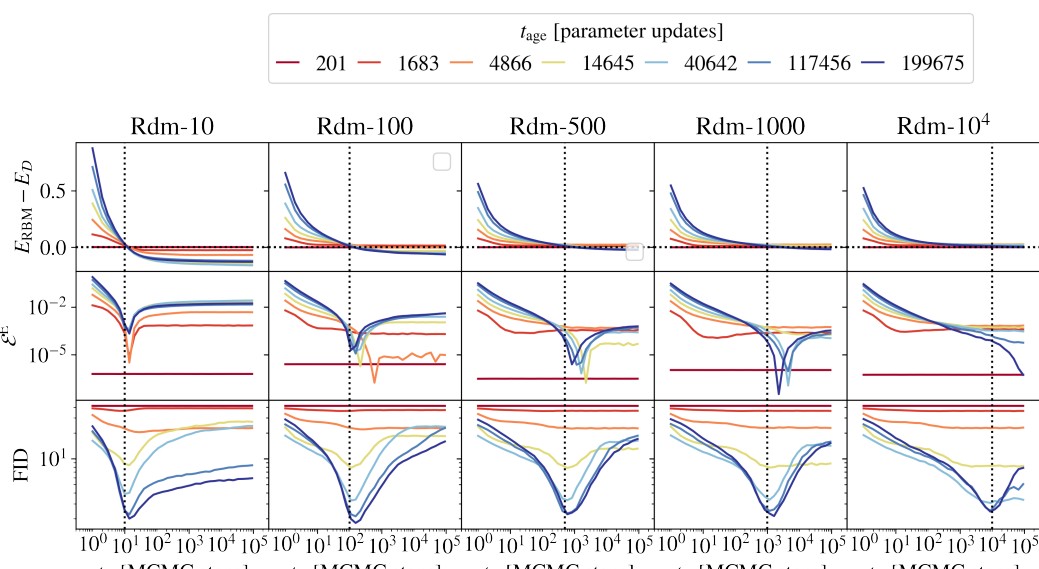

Figure 4: We show some extra estimators of the generated sample quality as function of the sampling time: the energy of the generated $E_{\mathrm{RBM}}$ and original $E_{\mathcal{D}}$ datasets, the error in the energy $\mathcal{E}^E$, and the FID score.

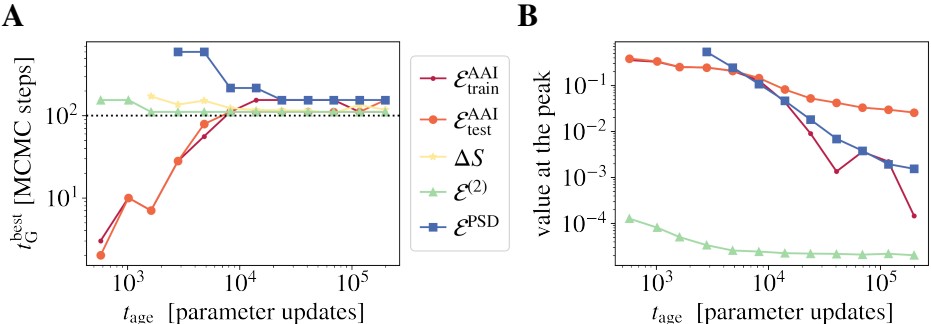

Figure 5: For RBMs trained with the Rdm-100 scheme, we show in **A**, the sampling time $t_{\mathrm{G}}$ at which we observe the best generation performance in the estimators shown in fig. 3 (that is, the position of the "best quality peak"), as function of the age of the RBM. In **B** we show the value reached by these estimators at the times shown in **A**, also versus the $t_{\mathrm{age}}$.

it in the $k = 10^4$ RBMs because poorly trained RBMs displayed extremely slow dynamics. In fact, these very slow relaxations prevented us to thermalize the system for most of the $t_{\mathrm{age}}$ studied in the low $k$ models. We can further elaborate this argument by studying the relaxations from configurations at different times. In particular, we consider two different RBMs (both trained with the Rdm-k scheme for the same number of updates $t_{\mathrm{age}} = 40620$): one using $k = 10$ and the other with $k = 10000$. Indeed, at that age, the Rdm-10000 RBM has learned the equilibrium model and the Rdm-10 RBM encodes a strongly OOE model, as discussed in the main-text. In order to characterize the relaxations of both models, we recompute the time-autocorrelation function discussed in the main-text. However, this time, we explicitly compare the magnetization at time $t + t_{\mathrm{w}}$, with the magnetizations at time $t_{\mathrm{w}}$,

$$\rho(t, t_{\mathrm{w}}) = \frac{C_m(t, t_{\mathrm{w}})}{C_m(t_{\mathrm{w}})}, \text{ where } C_m(t) = \frac{1}{N_v} \sum_i (m_i(t + t_{\mathrm{w}}) - \bar{m}_i)(m_i(t_{\mathrm{w}}) - \bar{m}_i) \tag{11}$$

We refer to $t_{\mathrm{w}}$ as the "waiting" time. Clearly, equilibrium is reached when $\rho(t, t_{\mathrm{w}})$ does not depend on $t_{\mathrm{w}}$ anymore. We show in fig.8-A and B, the relaxations of the Rdm-10 and Rdm-10000 models, respectively. Clearly, the relaxations of the Rdm-10 model are far slower than those of the Rdm-

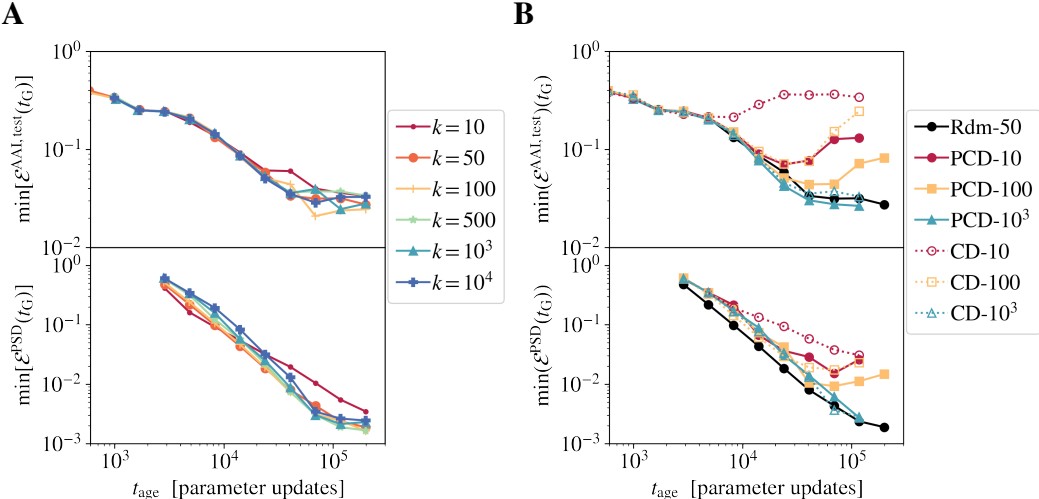

Figure 6: **A** We show the height of the best performance peak (for RBMs trained with the Rdm-k scheme) for the $\mathcal{E}^{\text{AAI}}$ (on the test set) and $\mathcal{E}^{\text{PSD}}$ estimators, as function of the machine's age for different values of $k$ (the same curves were already shown in fig. 5 in the case $k = 100$). We observe so little improvement of the generative performance with $k$ at the best performance peak. In **B**, we show the minimum of the values measured for these two estimators during the sampling of RBMs trained with the CD-k (dashed lines and empty symbols) and PCD-k (continuous lines and full symbols) scheme as function of $t_{\text{age}}$ and for several values of $k$. In this case, the quality of the generated samples improves notably with $k$ and even worsen over $t_{\text{age}}$ (something not observed in the Rdm scheme). For the sake of comparison, we also show the values obtained with the Rdm-50 scheme (in black, already shown in **A**), displaying remarkable better generation performances only comparable with the RBMs trained with $k = 1000$.

10000. Showing the former, actually, forever aging effects extremely similar to those observed in spin glasses.

### 3.3 CD-k scheme

In the CD case on fig. 10, we see that for small $k$, the system enters quickly in the OOE regime. However in that case, the interaction between the initial condition and the dynamics lead to a behavior hard to predict: the peak of best performance, if it exists, moves with the age of the machine, and the general performance are not good. When $k = 1000$ we observe quite a good performance. This is due to the fact that at the beginning of the learning this number of MCMC steps is sufficiently high to decorrelate from the initial condition. As $t_{\text{age}}$ increases, the dataset is presumably a better initial condition than taking a random one, and the length of the MC chain is probably enough to thermalise correctly. This agrees with the fact that $k = 1000$ is comparable to the highest values of the mixing time $t_\alpha$ shown in the main-text. We should still remark that in that case, $t_{\text{G}}$ needs to be very large if one wants to generate equilibrium configurations during the sampling.

In fig. 9 we illustrate the behavior of CD-100 both when sampling using the dataset as initial condition or starting from random ones. It is clear that the configurations visited when starting from random almost never reach a digit (even with $k = 100$). When initialized at the dataset, and as explained before, the system explore nearby configurations (we can see that the chains do not decorrelate from the original digit) until it manages forget the initial configuration and starts to sample spurious states (which are definitely not numbers).

### 3.4 PCD-k scheme

Let us finish with the PCD case. We give a glance of the sample generation process of a RBM trained with PCD-100 in fig. 12. We have considered two possible starts for the MC chain: random and dataset, and for both cases, it seems clear that we need at least 10000 sampling MCMC steps to start to sample good digits that are also uncorrelated initialization of the chains. We quantify these

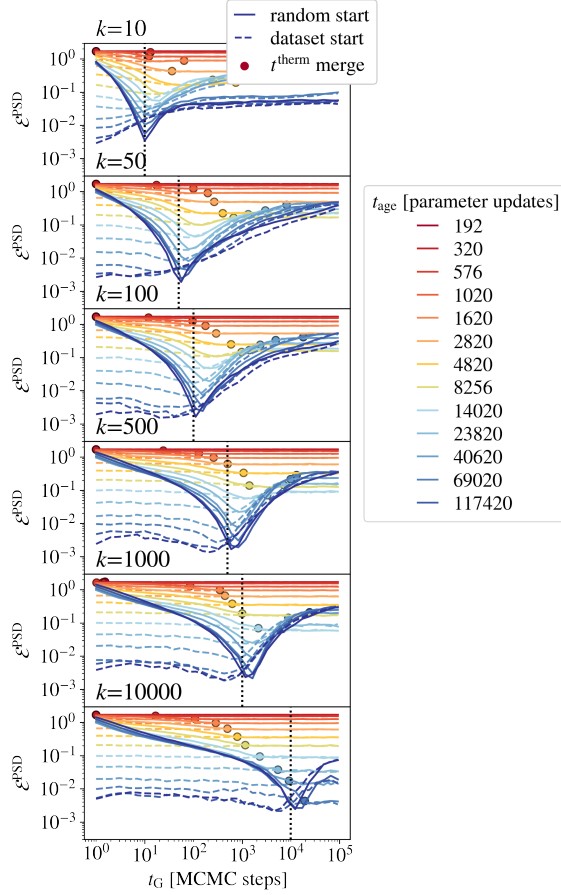

Figure 7: Evolution of $\mathcal{E}^{\mathrm{PSD}}$ during the sampling of RBMs trained with MNIST data and with the Rdm-k scheme for different values of $k$ (different cells) and different $t_{\mathrm{ages}}$ (coded by colors). In continuous lines, we show the measures obtained with runs initialized at random, and in dashed lines, those from runs initialized at the dataset. We highlight with a circle, the point at which both measures merge to the same constant curve. The $k = 10^4$ curve was included in fig. 3-B of the main-text.

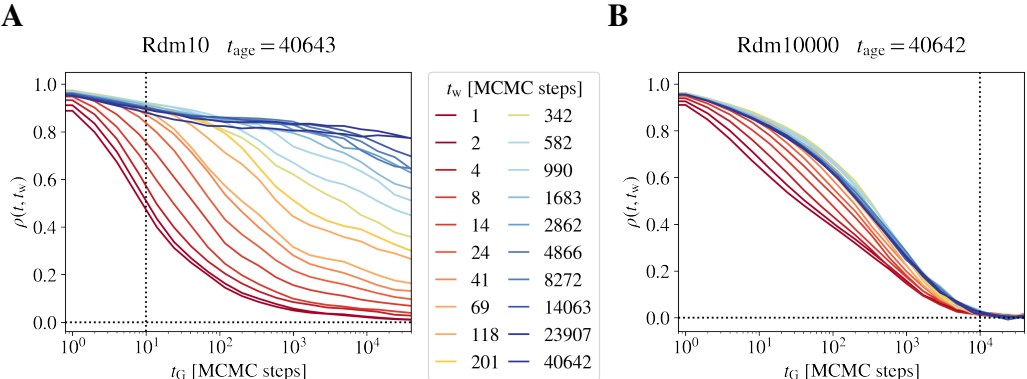

Figure 8: We show the two-time-autocorrelation function of the magnetizations, $\rho(t, t_{\mathrm{w}})$, defined in Eq. 11, during the sampling of two RBMs: either trained with Rdm-10 (in **A**) or Rdm-10000 (in **B**). Both RBMs were trained for a fixed number of updates $t_{\mathrm{age}} = 40642$ (or 40643) and we code by colors the value of the waiting time $t_{\mathrm{w}}$ used to compute $\rho(t, t_{\mathrm{w}})$. In **A**, the longer we wait, the slower the relaxation for all the $t_{\mathrm{w}}$ considered. In **B**, curves above $t_{\mathrm{w}} > 500$ converge to the same relaxational curve, which coincides with the one showed for that age in fig.3-A of the main-text.

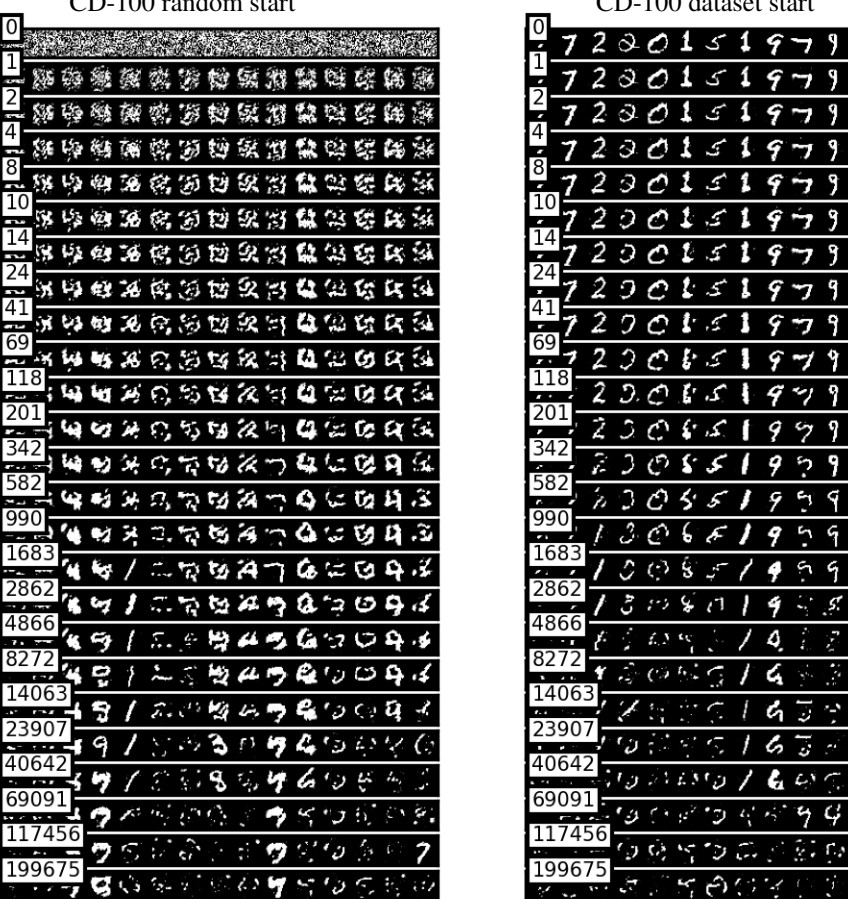

Figure 9: Same experiment shown in fig. 2, but this time sampling an RBM trained with the CD-100 scheme for $t_{\text{age}} = 199876$ parameter updates. Even with $k = 100$, RBMs trained with the CD-$k$ scheme are not able to create digits from scratch, as shown in the random initialization Markov chains. If instead, the chains are initialized at the dataset, they remain close to the original digit for at least $t \sim 3000$ MCMC steps, without decorrelating from the initial condition. For larger times, the machine does not generate digits anymore.

observations by looking at the evolution of the quality observables with the sampling time (see fig. 11). In this case, the PCD approximation scheme does a good job, the permanent chain manages to keep the RBM at quasi-equilibrium during all the learning, which is consistent with the observation that RBM's parameters change really slowly during the learning. In fact, we see almost no memory effects in the dynamics, even for small $k$. Yet, it is important to remark that the time needed to obtain equilibrated samples from scratch is quite high. In addition, the quality of the best generated samples using this training scheme are significantly worse than those obtained in the OOE regime of Rdm-$k$ Also the AAI score on the test set is not as good as for the Rdm-$k$ scheme unless we reach MCMC chains of $k = 1000$, as shown in fig. 6-B.

### 3.5 Comparisons between different training schemes

We have shown that different training schemes generate RBMs that display different generation sampling dynamics (when trying to create digits from scratch just through the sampling of their Gibbs equilibrium distribution). Indeed, the evolution of the quality estimators for Rdm-$k$, CD-$k$ and PCD-$k$ are shown in Figs. 3, 10 and 11 respectively. For the Rdm case, we see that that by increasing $k$ we reach equilibrium for larger $t_{\text{age}}$, but eventually we end up in the OOE regime as the mixing time keeps increasing with the number of LL gradient updates. For CD, it seems that for $k = 10, 100$, the RBM enters in an uncontrolled OOE regime, while when $k = 1000$, the machine

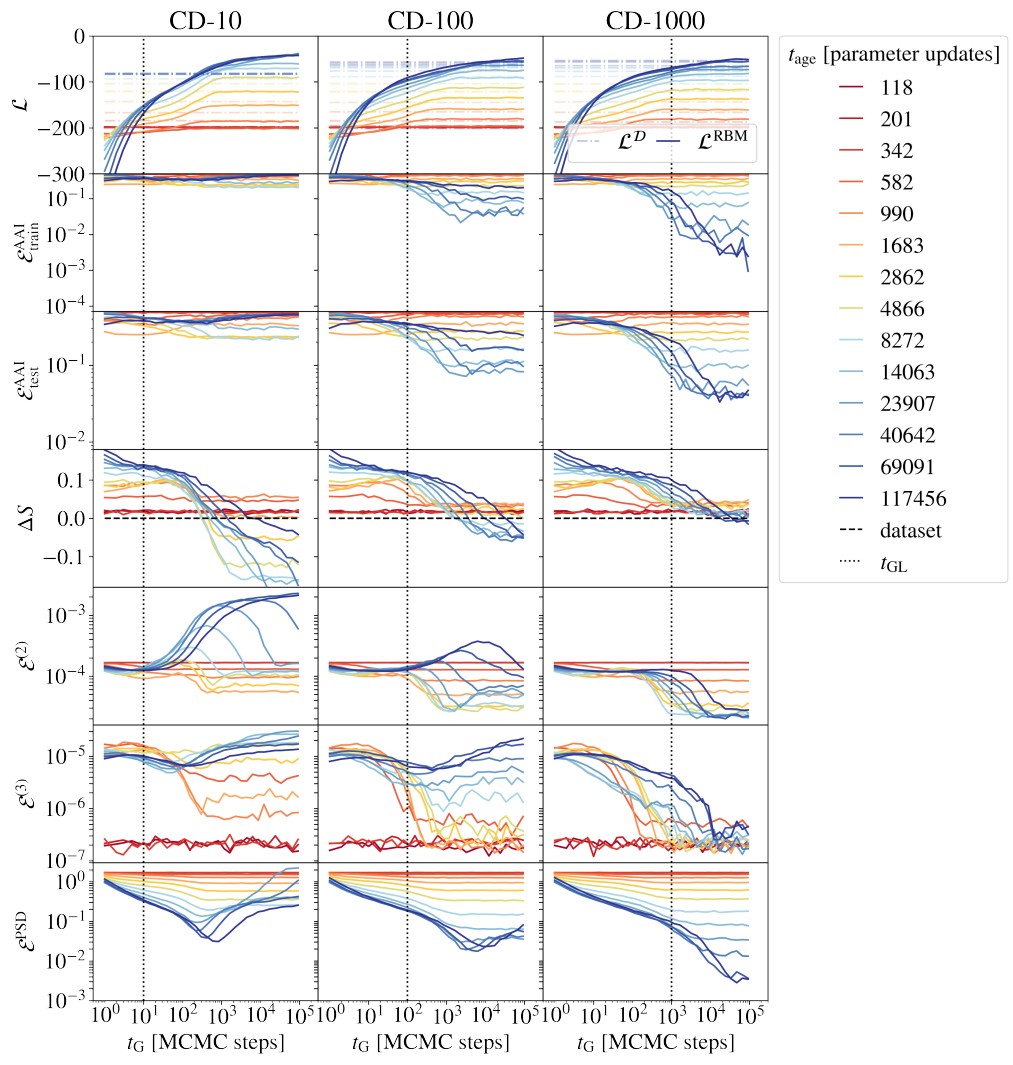

Figure 10: Evolution of the quality estimators along the sample generation for MNIST database for RBMs trained using the CD-k approach and with different values of $k$.

seems to have a nice equilibrium behavior (to confirm such behavior one should quantify the mixing time). Finally, in PCD, when $k \geq 100$ we observe that the machine ends up in a good equilibrium regime where the quality of the generated samples increase with $k$. Again, the mixing time should be controlled to confirm this conclusion. Still, we can also observe that the best generation performance is not necessarily reached on this equilibrium regime. Indeed, we can observe in fig. 6 that RBMs running on the OOE regime generates far better samples than CD or PCD for short values of $k$, and very similar for large $k$. In fact, in fig. 6-B, we show the minimum value measured in two quality estimators as function of $t_{\mathrm{age}}$ for the different schemes, showing that the Rdm-50 runs create samples as good as PCD-1000 ones at long $t_{\mathrm{age}}$ (despite being the training of the former 20 times faster). For shorter values of $t_{\mathrm{age}}$, the quality of the samples generated by the Rdm-50 RBMs, always that of overpasses CD or PCD-1000.

## 4 Results for a Human Genome dataset

In this case we considered the population genetics' dataset of Ref. [7]. This dataset corresponds to a sub-part of the genome of a population of 5008 individuals, for which the alteration or not of a given gene is indicated by a $\{0, 1\}$ variable. A total of 805 genes are reported. For our study, the

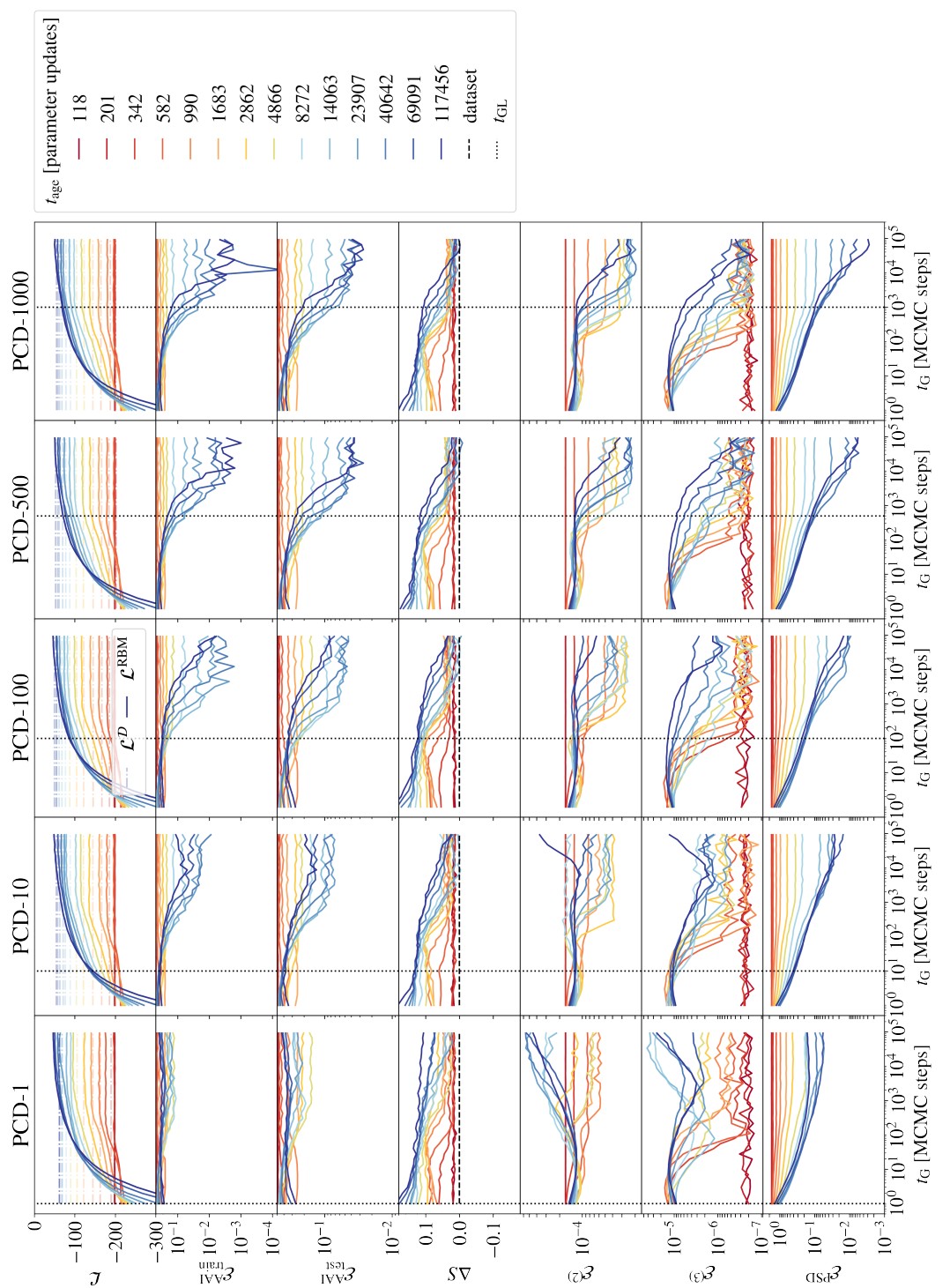

Figure 11: Evolution of the quality estimators along the sample generation for MNIST database for RBMs trained using the PCD-k approach and for different values of $k$.

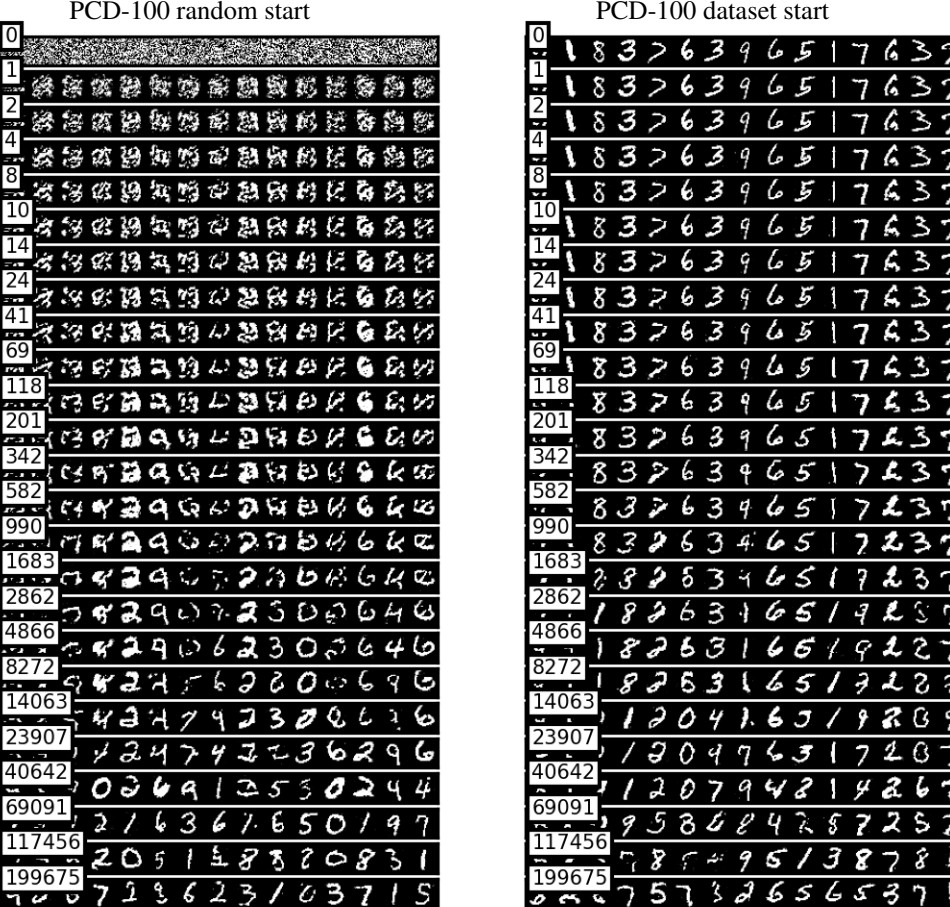

Figure 12: We show some examples of the images we obtain at different steps during the data generation sampling of an RBM trained with the PCD-100 scheme for $t_{\text{age}} = 199876$ parameter updates following the same procedure discussed in fig. 9.

dataset was separated into $4500$ samples for the train set and $500$ for the test set. For this dataset, the following parameters of the RBM were used:

- Number of hidden nodes: $N_{\text{h}} = 100$
- Learning rate: $\alpha = 0.01$
- Minibatch size: $n_{mb} = 250$
- For the indicators, we used all the training set $N_s = 4500$ (and only $500$ for measurement involving the test set).
- the rest is the same as for MNIST.

We observe for this dataset qualitatively the same behavior as for MNIST. We show on fig. 13 a figure similar to fig. 3 in the main text for this dataset. We observe, again, two clear regimes: the equilibrium and the OOE. The equilibrium regime is observed for almost all the RBMs trained with $t_{\text{age}} \lesssim 3000$ updates (except for the Rdm-10 RBM), and the OOE regime, for older machines. It is quite remarkable that the samples generated by RBMs of age $t_{\text{age}} = 2862$, reproduce quite well the dataset's 2 and 3 points-correlations (as shown in the $\mathcal{E}^{(2)}$ and the $\mathcal{E}^{(3)}$ observables), but display quite poor values for $\mathcal{E}^{\text{AAI}}$. The fact that all the RBMs with $k > 10$ reproduce essentially the same curves with time, tells us that the relaxation time is rather low below this age, surely over 10 MCMC steps and below 50. However, beyond this age, the mixing time grows drastically, clearly overpassing the $10^4$ steps since none of our RBMs describe the equilibrium dynamics for any of the blue lines. Yet, even in the OOE regime, we are still able to generate very good samples (as reflected

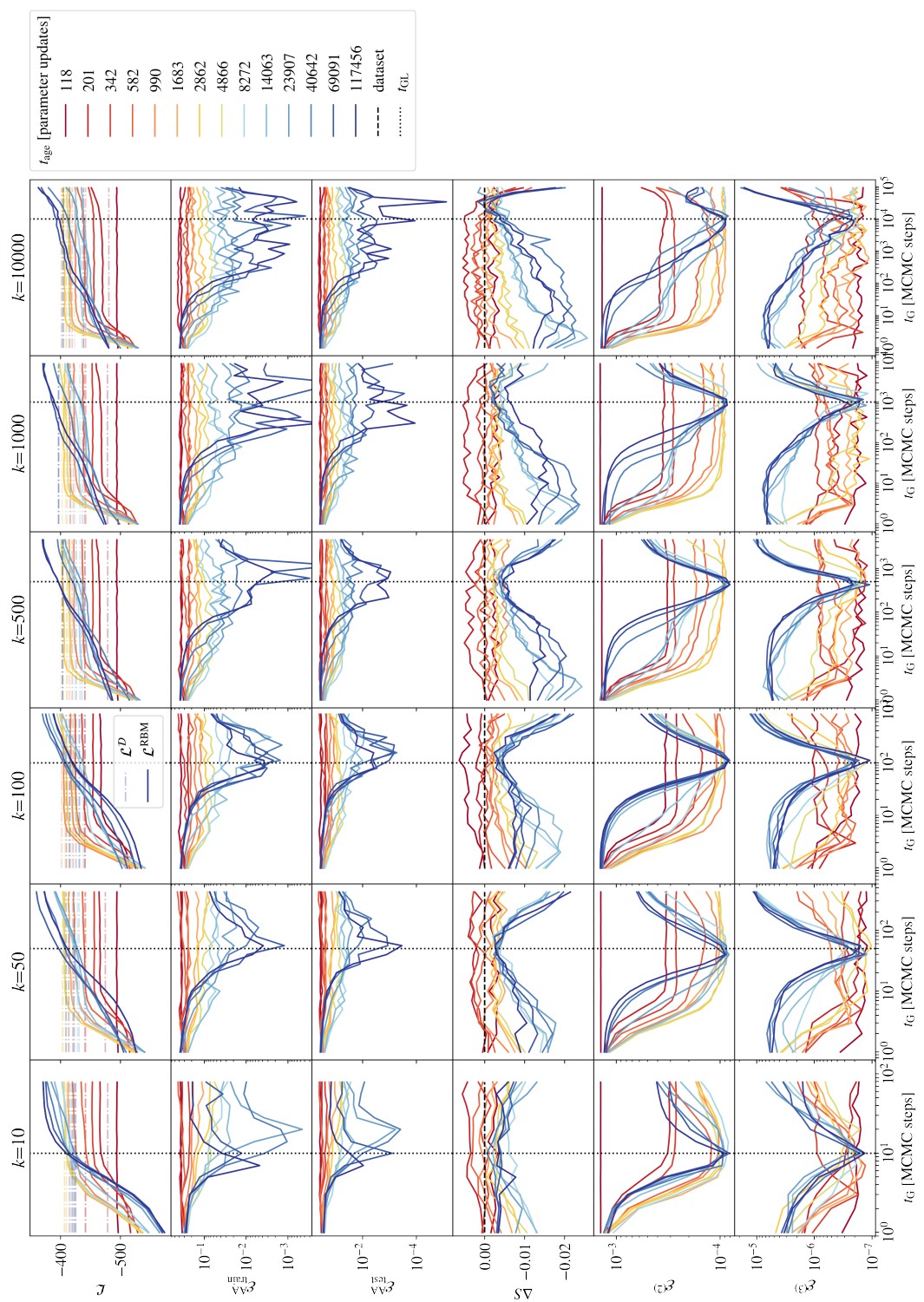

Figure 13: Evolution of the quality estimators during the sampling of the dataset of population genetics for RBMs trained using the Rdm scheme and for various values of $k = 10, 50, 100, 500, 1000, 10000$. We clearly see the same behavior where the best performance are obtained for $t_G \sim k$. The system never manage to equilibrate, even for $k = 10000$.

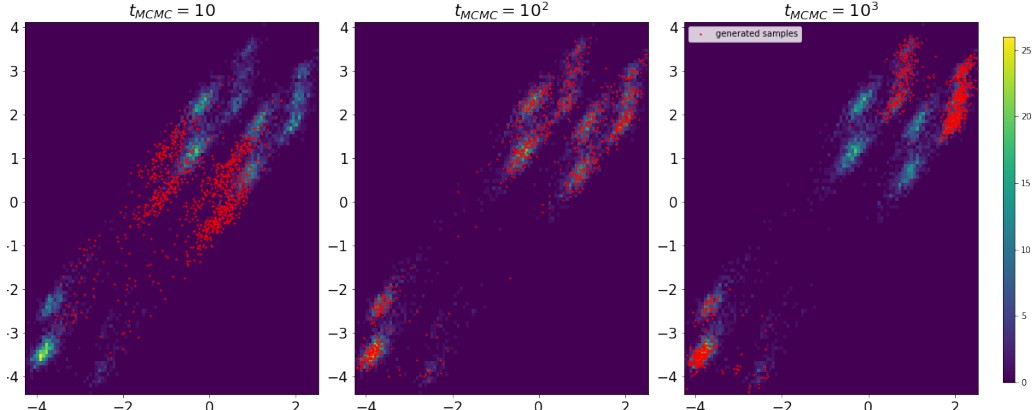

Figure 14: All three images: histogram (as heatmap) of the scatter plot along the first two eigenvectors of $w$ of the dataset. From left to right: the red dots correspond to the scatter plot of 1000 samples generated by the RBM at respectively at $t_{\mathrm{G}} = 10, 10^2, 10^3$ Monte Carlo steps. The RBM has been trained using the Rdm-100 scheme, and the best match between the projections of the original and generated dataset are observed precisely at $t_{\mathrm{G}} \sim k = 100$.

in all the quality observables) as long as we limit the generation sampling to $t_{\mathrm{G}} \sim k$ MCMC steps for each RBM (just as observed in the MNIST's case).

Finally, we believe that this sudden and brutal growth of the mixing time with $t_{\mathrm{age}}$ is related to the structure of this dataset: the MCMC simulations get trapped in some relative maxima of the LL, and it takes extremely large times to escape from them. To illustrate this, we can project the samples either on the first principal directions or, on the left-eigenvectors of $w_{ia}$ as discussed in [8]. We show in fig. 14, the projection of the dataset along the directions of the first two eigenvectors of the dataset (being each the horizontal and vertical axis, respectively), and the projection of the generated samples at 3 different sampling times. The original dataset is shown as a heatmap of the point density, and the generated dataset as red dots. We clearly show that at $t_G = k$, the generated samples provide a very good cover of the dataset density. At the opposite, after very few updates, it didn't spread correctly, and after too many updates, the samples cluster into a small subset of the data density distribution.

## 5 Results for CelebA

This dataset corresponds to a total of 30000 faces of celebrity, in color and potentially in high-resolution. In our case, we first downsized the dataset to a resolution of $128 \times 128$ in order to reduce the computational time of the learning part. Since RBM are more suited to handle binary entries, we then clipped the images's pixels at a value of 0.3. For our study, we only considere a train set of 30000 samples. For this dataset, the following parameters of the RBM were used:

- Number of hidden nodes: $N_{\mathrm{h}} = 5000$
- Learning rate: $\alpha = 0.01$
- Minibatch size: $n_{mb} = 500$
- For the indicators, we used $N_s = 1000$ samples.
- the rest is the same as for MNIST.

We observe for this dataset qualitatively the same behavior as for MNIST, even though it is clearly much harder to learn. Due to the learning time, we will show only the results for $k = 50, 100, 500$. We show on fig. 15 the evolution of the sampling for our indicators and for many values of $t_{\mathrm{age}}$. We also show on fig 16 sampling obtained for thr Rdm-100 machine for several values of $t_G$, demonstrating the RBM can indeed learn quite complex dataset.

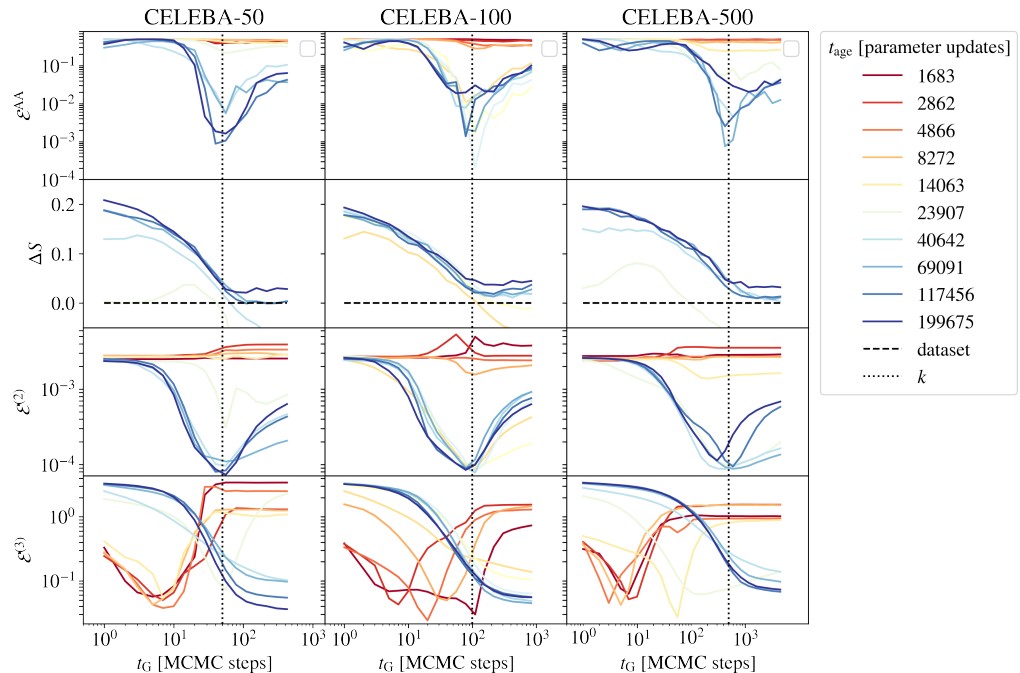

Figure 15: Evolution of the quality estimators during the generation sampling of CELEBA images, for RBMs of different ages ($t_{\mathrm{age}}$ coded by colors), and trained using the Rdm-k scheme with $k = 50,\ 100$ and $500$ in different columns. The best generation performance is also observed clearly around $t_{\mathrm{G}} \sim k$, except for the PSD observable which is not able to characterize properly these kinds of images. For this case, we remove the indicator on the third moment since it was too computationally demanding.

# 6    Results for FashionMNIST

The FashionMNIST[9] dataset corresponds to a total of $60000$ images of $28 \times 28$ pixels in grayscale. The dataset represents various classes of clothers (shirts, pants, ...). For this dataset, we train the RBM on input rescaled within $[0, 1]$ but we didn't discretize the values as we did for MNIST. The following parameters were used for the training:

- Number of hidden nodes: $N_{\mathrm{h}} = 1000$
- Learning rate: $\alpha = 0.01$
- Minibatch size: $n_{mb} = 500$
- For the indicators, we used $N_s = 1000$ samples.
- During the training we used $k = 100$ Gibbs steps.
- the rest is the same as for MNIST.

We show on fig. 17 a subset of the dataset together with the generated images of the RBM. Then, we show on fig. 19 the results of several indicators for this case, showing again the characteristic non-equilibrium best-performance peak around $t_{\mathrm{G}} \approx k$. Finally, we show visually how the generated images evolve with the sampling time, up to a very long number of MCMC steps in fig. 18.

# 7    Results for Caltech101 Silhouettes

This dataset corresponds to a total of $8671$ images of $28 \times 28$ pixels in black and one, corresponding to the silhouette of various objects.. The dataset is made of binary input from which not pre-treatment was made. The following parameters were used for the training:

- Number of hidden nodes: $N_{\mathrm{h}} = 500$

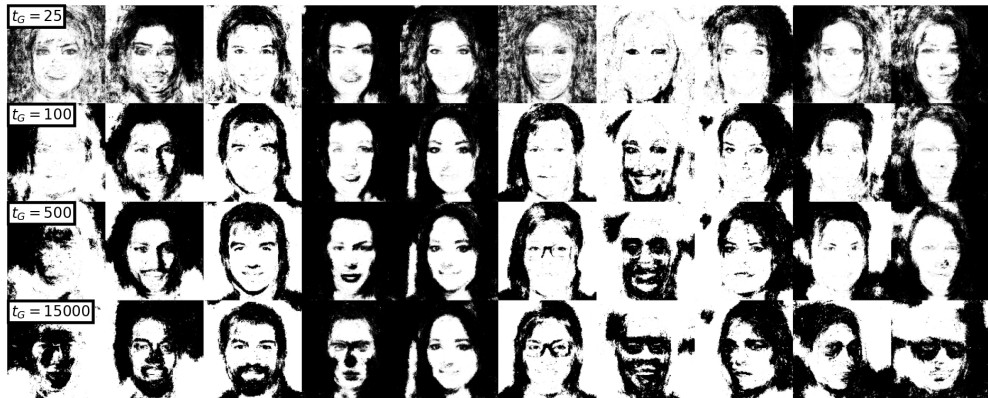

Figure 16: Faces generated by the RBMs starting from random initial conditions, trained with Rdm-100 and up to $t_{age} = 360000$. Each row represent a fixed number of sampling steps. We can see at the beginning that very quickly the MC chains is able to generate very blurry faces. Then, when $t_G \sim O(100)$ it generates quite decent faces. Later, the chains seem to be biased toward more obscure version of the faces.

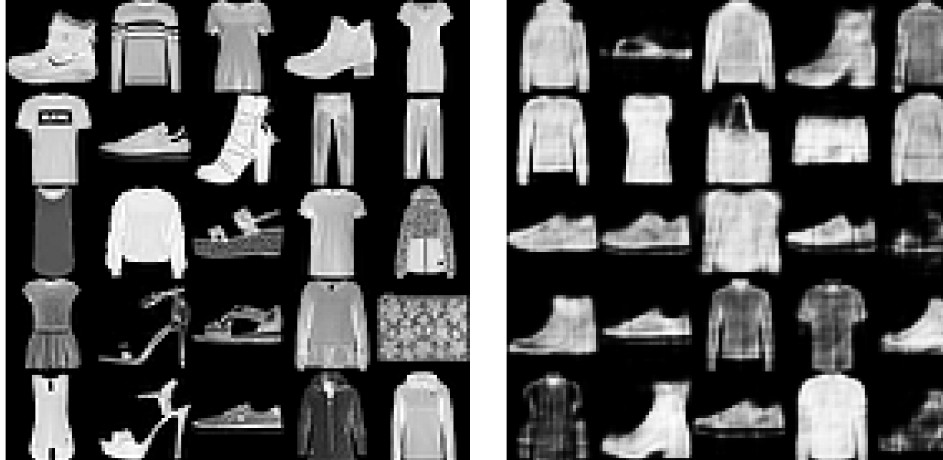

Figure 17: **Left:** A subset of FashionMNIST showing 25 random samples. **Right:** 25 images generated from the RBM trained of the FMNIST dataset. The RBM was trained up to $t_{age} = 200k$ and extracted at $t_G = k = 100$

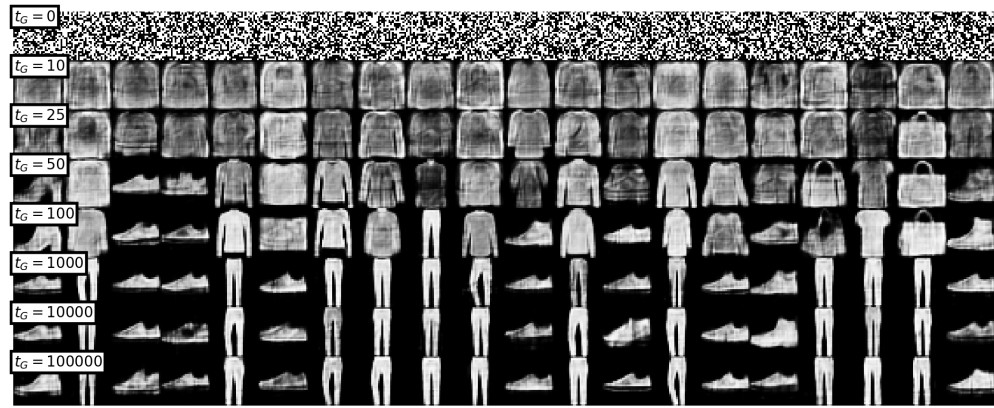

Figure 18: Various samples obtained from the RBM at different values of $t_G$. We see that when $t_G = k$ we obtain quite good samples while for large sampling time the generated images over-represent pants and shoes.

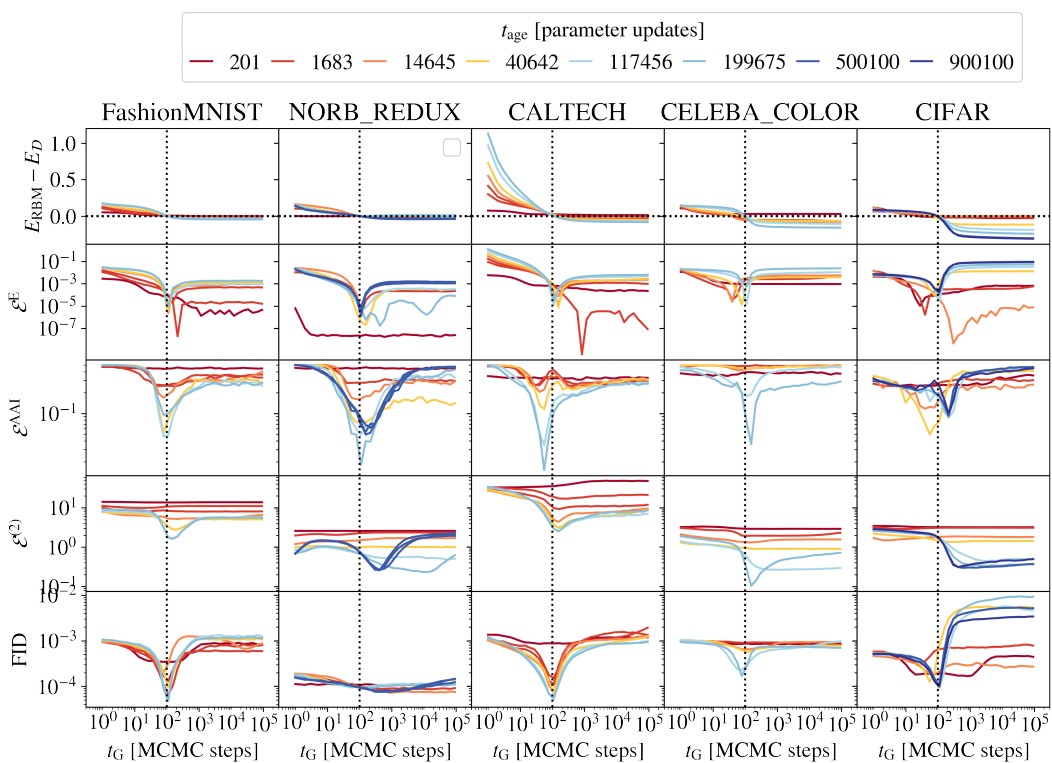

Figure 19: We show the evolution of several quality estimators for RBMs trained with $k = 100$ in 5 different datasets: FashionMNIST, NORB REDUX, CALTECH101 Silhouettes, CELEBA in colour and CIFAR. Each dataset is shown in a different column.

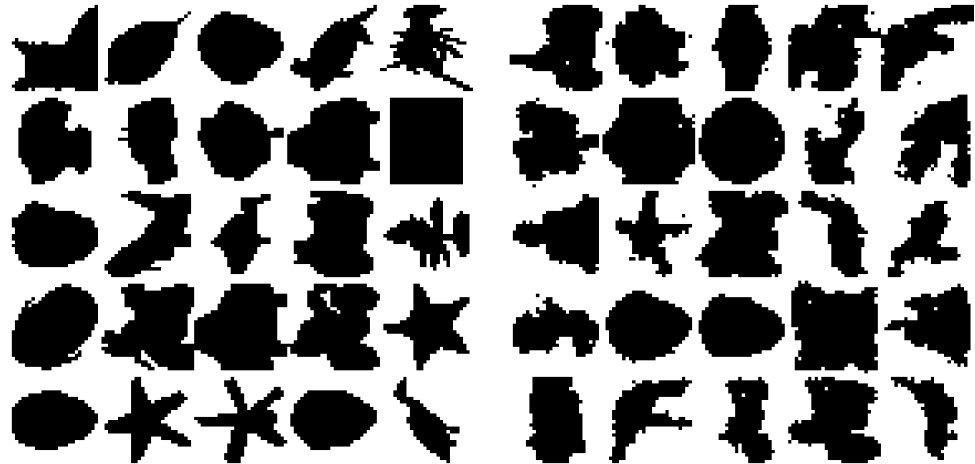

Figure 20: **Left:** A subset of FashionMNIST showing 25 random samples. **Right:** 25 images generated from the RBM trained of the FMNIST dataset. The RBM was trained up to $t_{\text{age}} \approx 200k$ and extracted at $t_G = k = 100$

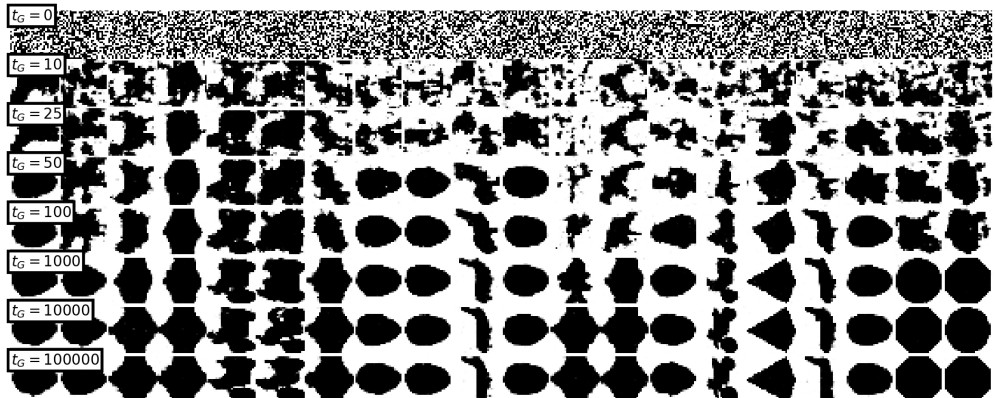

Figure 21: Various samples obtained from the RBM at different values of $t_G$. We see that when $t_G = k$ we obtain quite good samples while for large sampling time the generated images over-represent some shapes.

- Learning rate: $\alpha = 0.01$
- Minibatch size: $n_{mb} = 500$
- For the indicators, we used $N_s = 1000$ samples.
- During the training we used $k = 100$ Gibbs steps.
- the rest is the same as for MNIST.

We show on fig. 20 a subset of the dataset together with the generated images of the RBM. Then, we show on fig. 19 the results of several indicators for this case showing again the non-equilibrium regime when $t_G \approx k$. Finally, we show how the samples evolves on fig. 21 when we let the sampling for a large number of MC steps.

## 8 Results for smallNORB dataset

This dataset [10] corresponds to a total of 48600 images of $96 \times 96$ pixels in grayshade dividing in two sets of equals size (train and test). The images correspond to pictures of small toy taken under

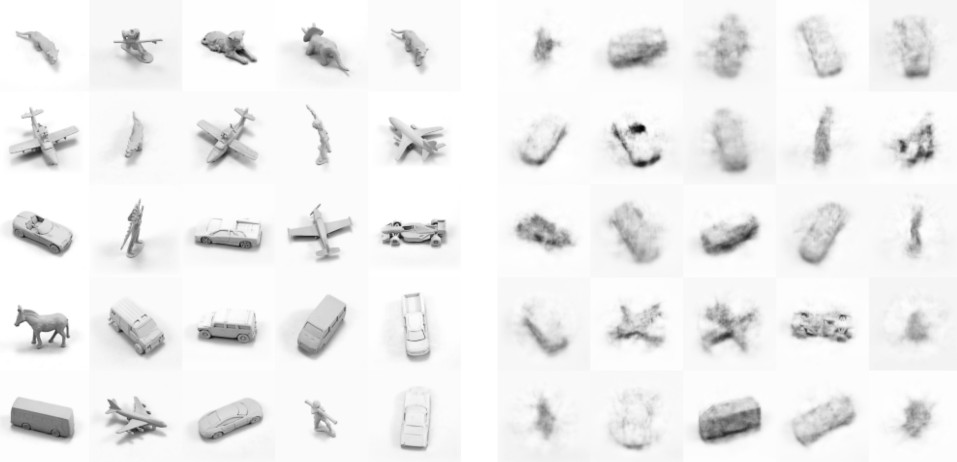

Figure 22: **Left:** A subset of FashionMNIST showing 25 random samples. **Right:** 25 images generated from the RBM trained of the FMNIST dataset. The RBM was trained up to $t_{\mathrm{age}} \approx 200k$ and extracted at $t_{\mathrm{G}} = k = 100$

different condition (light, angle ...). For the purpose of our work and to simplify the learning, we gather all the possible samples (amongst the train and test sets) selecting only one type of lighting. This reduced the dataset to 16300 images, which we refer to as NORB_REDUX. The RBM was trained on the gray shaded images using the following parameters:

- Number of hidden nodes: $N_{\mathrm{h}} = 2000$
- Learning rate: $\alpha = 0.01$
- Minibatch size: $n_{mb} = 500$
- For the indicators, we used $N_s = 1000$ samples.
- During the training we used $k = 100$ Gibbs steps.
- the rest is the same as for MNIST.

We show on fig. 22 a subset of the dataset together with the generated images of the RBM. Then, we show on fig. 19 the results of several indicators for this case showing again the non-equilibrium regime when $t_{\mathrm{G}} \approx k$. Finally, we show how the samples evolves on fig. 23 when we let the sampling for a large number of MC steps.

## 9   Results for CIFAR10 dataset

This dataset [11] corresponds to a total of 60k images of 10 classes of objects in color. The resolution of each image is $32 \times 32$. Since the dataset is in color, the total number of input is $3 \times 32^2$. The images correspond to pictures of celebrity. The RBM was trained on the colored images using the following parameters:

- Number of hidden nodes: $N_{\mathrm{h}} = 1000$
- Learning rate: $\alpha = 0.01$
- Minibatch size: $n_{mb} = 500$
- For the indicators, we used $N_s = 1000$ samples.
- During the training we used $k = 100$ Gibbs steps.
- the rest is the same as for MNIST.

We show on fig. 24 a subset of the dataset together with the generated images of the RBM. Then, we show on fig. 19 the results of several indicators for this case showing again the non-equilibrium

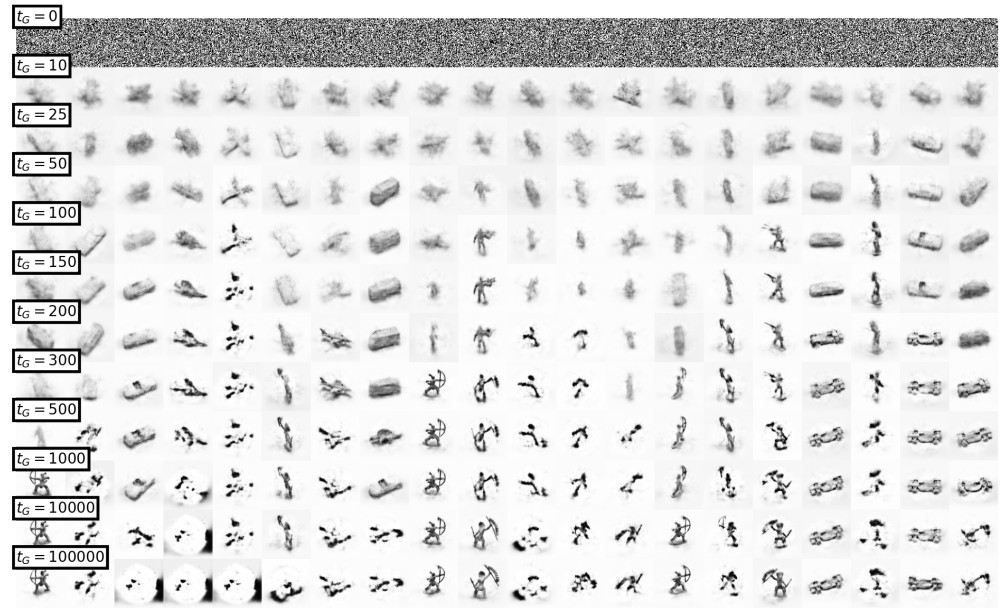

Figure 23: Various samples obtained from the RBM at different values of $t_G$. We see that when $t_G \approx k$ we obtain quite good samples while for large sampling time, the generated images are sometime basd or only of a particular type.

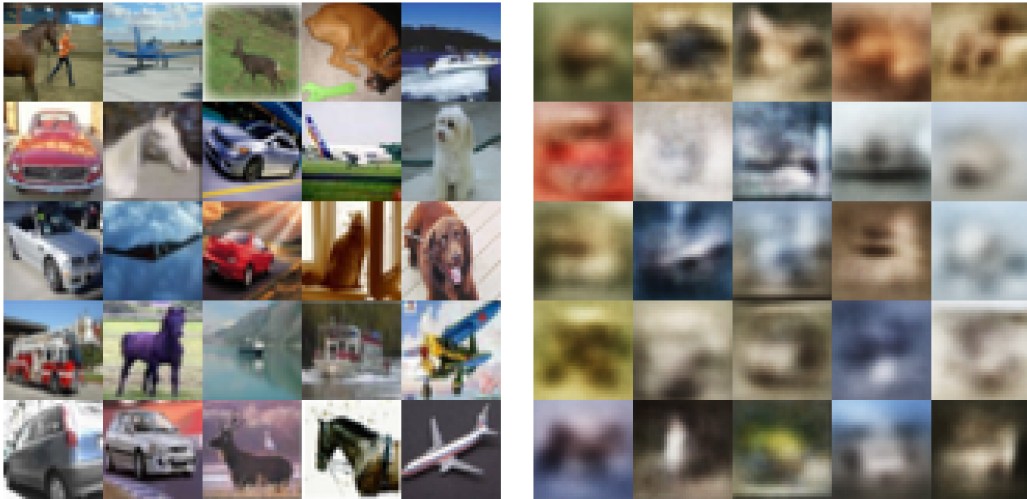

Figure 24: **Left:** A subset of CIFAR10 showing 25 random samples. **Right:** 25 images generated from the RBM trained of the CIFAR10 dataset. The RBM was trained up to $t_{age} \approx 900k$ and extracted at $t_G = k = 100$

regime when $t_G \approx k$. Finally, we show how the samples evolves on fig. 25 when we let the sampling for many MC steps.

## 10   Results for CelebA dataset

This dataset[12] corresponds to a total of more than 200k images of celebrity. For the purpose of our work, we retain only 12000 images, which we downsample to $64 \times 64$ pixels. Since the dataset

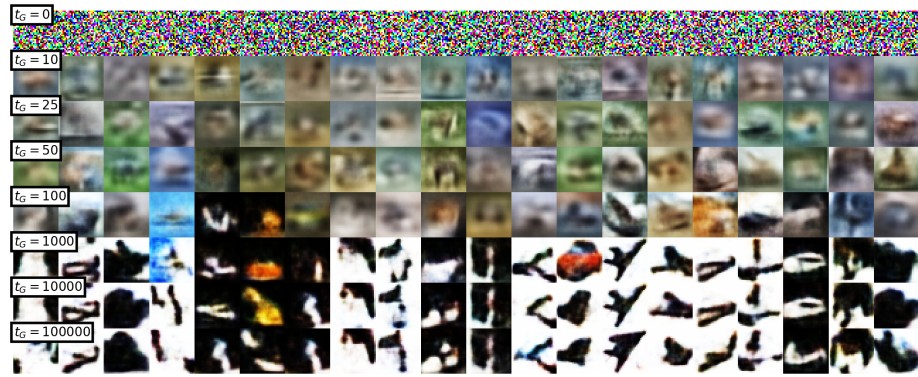

Figure 25: Various samples obtained from the RBM at different values of $t_G$. We see that when $t_G = k = 100$ we obtain samples with great variety while for large sampling time the generated images get a very biased look.

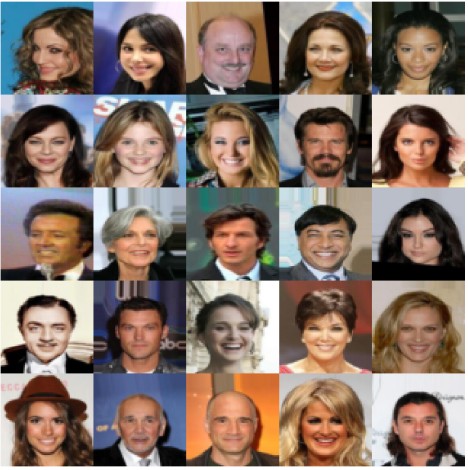 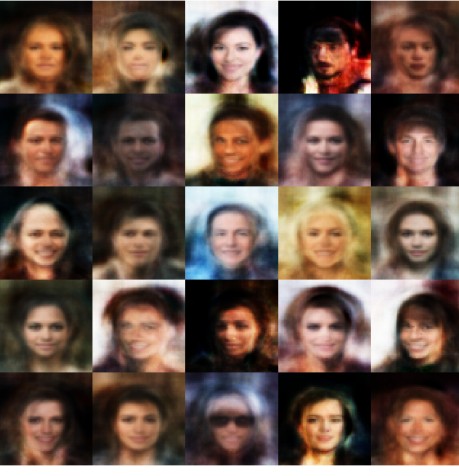

Figure 26: **Left:** A subset of CELEBA showing 25 random samples. **Right:** 25 images generated from the RBM trained of the CELEBA dataset. The RBM was trained up to $t_{age} \approx 100k$ and extracted at $t_G = k = 100$.

is in color, the total number of input is $3 \times 64^2$. The images correspond to pictures of celebrity. The RBM was trained on the grayshaded images using the following parameters:

- Number of hidden nodes: $N_h = 5000$
- Learning rate: $\alpha = 0.01$
- Minibatch size: $n_{mb} = 500$
- For the indicators, we used $N_s = 1000$ samples.
- During the training we used $k = 100$ Gibbs steps.
- the rest is the same as for MNIST.

We show on fig. 26 a subset of the dataset together with the generated images of the RBM. Then, we show on fig. 19 the results of several indicators for this case showing again the non-equilibrium regime when $t_G \approx k$. Finally, we show how the samples evolves on fig. 27 when we let the sampling for many MC steps.

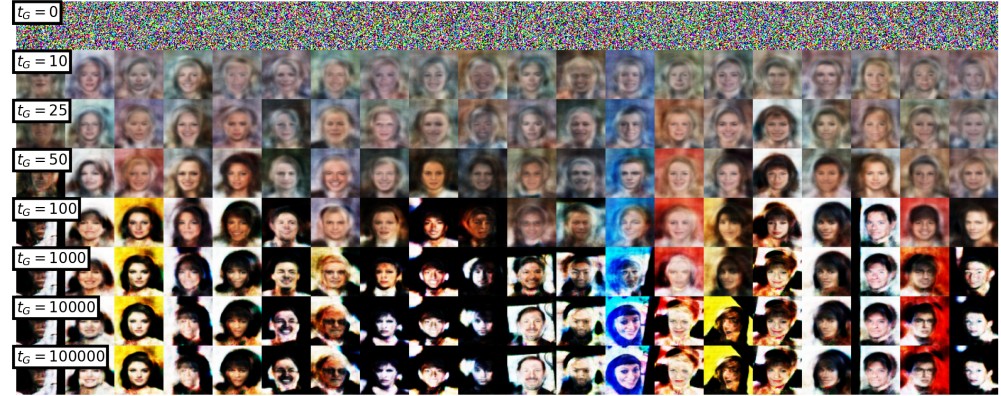

Figure 27: Various samples obtained from the RBM at different values of $t_G$. We see that when $t_G = k = 100$ we obtain quite good samples while for large sampling time the generated images over-represent more regulare images.