# OpenReview forum: "Equilibrium and non-Equilibrium regimes in the learning of Restricted Boltzmann Machines"
_NeurIPS.cc/2021/Conference — NeurIPS 2021 Poster_

### Official Review · Reviewer_g5Pg · 2021-07-15

**Rating:** 6
**Confidence:** 5

**Summary:**

The authors demonstrate that many RBM models are learned in an unexpected out-of-equilibrium regime where the distribution of short run MCMC samples used to update the model differs greatly from the long run equilibrium distribution of the model. The claims are supported with diagnostics to measure out-of-equilibrium behavior and visualizations of samples after across MCMC updates.

**Limitations And Societal Impact:**

The authors have adequately addressed limitations and impact of their work.

**Main Review:**

The main objective of this paper is to demonstrate that RBM training has an unexpected but widespread behavior where samples generated by short run sampling using common initialization techniques like CD and PCD are typically out of equilibrium. Despite this, learning can remain stable and generate short run images that are very similar to data images, and the apparent success of learning a generative model can lead to the mistaken conclusion that the RBM has learned a valid equilibrium distribution. The authors question common practice of evaluating RBM's exclusively using short-run samples, which are not indicative of the steady-state, or negative samples from the end of training, which might not even be configurations which are attainable by the model once the weights are frozen. To examine the equilibrium distribution of RBM's, the authors use long run MCMC samples and find that equilibrium states are poor samples across many learning configurations. The points raised by the authors are crucial from the perspective of rigorous probabilistic learning of an RBM and this work sheds light on important but under-recognized phenomena in generative modeling.

The main weakness of this work is that the authors do not sufficiently address the related work [a] which covers many of the same ideas. The work [a] identifies a convergent and non-convergent learning phase for the Energy-Based Model (EBM) that directly correspond to the equilibrium and out-of-equilibrium phases identified by the authors for RBM training. It appears that this phenomenon presented by the authors occurs in energy models with different kinds of potentials. Given the high degree of similarity between these two works, I recommend against acceptance until a discussion of the connections to [a] is made clear by the authors in a revised version.

The authors present several diagnostics to identify equilibrium and out-of-equilibrium behavior. The most convincing diagnostics to me were the auto-correlation measures over the course of training. The other diagnostics seemed to be somewhat ad-hoc but nonetheless they appear to be useful indicators. One diagnostic that that authors did not include but possibly should is a measurement of the energy of the visible states over the course of training. Typically one would expect that the energy of short run MCMC samples should be about the same as data samples. The work [a] shows that when the short run samples are out of equilibrium, then long-run samples typically have much lower energy than short run or data samples. Successful learning should occur when short run, long run, and data energy values remain in the same spectrum, and it might be possible to diagnose out of equilibrium behavior for RBMs by examining whether long run visible energy values are in the same spectrum as short run and/or data samples.

The scale and complexity of the experiments is somewhat limited because the authors exclusively examine binary pixel states rather than continuous and multi-channel pixels. The authors clearly demonstrate that out-of-equilibrium learning occurs for RBM models. I am a little uncertain over the author's exact conclusion regarding the method for learning an equilibrated model. The abstract claims that increasingly large k can be used to learn a valid equilibrium, but experiments with Rmd-k show even for k=10,000, non-equilibrium effects appear after around 100k MCMC updates. On the other hand, PCD learns a more reasonable equilibrium even for smaller k, which is consistent with what is reported in [a]. It would be good for the authors to clarify strategies for learning a correct equilibrium.

**Overall Summary**: This work identifies an important and under-recognized outcome of out-of-equilibrium RBM learning. Bringing attention to this phenomenon is useful and valuable. The major weakness of this work is that it does not include a related work [a] that presents very similar findings. The experimental complexity could also be increased by using continuous pixels and color images. I recommend against accepting this work until relations to prior work are made clear. I am willing to change my score based on author feedback and revisions.

[a] On the anatomy of MCMC-based maximum likelihood learning of energy-based models. E. Nijkamp, M. Hill, T. Han, Y.N. Wu, S.C. Zhu. AAAI 2020. https://arxiv.org/abs/1903.12370

**Time Spent Reviewing:**

5 hours

---

> ### Author Response · Authors · 2021-08-09
> **Answer to ref g5Pg (1/2)**
>
> Before beginning with the discussion, we want to thank the referee for his/her careful reading of the manuscript and his/her insights and valuable constructive comments about how to improve the manuscript and to frame it in the existing literature of EBMs. We are particularly happy to see that the referee found the results interesting, important and sound.
>
> 1) **Missing references.** We really want to thank the referee for mentioning Ref. [a] (and with it, its companion work [b]) that had skipped our radar, and are, indeed, very relevant for our work. The empirical results discussed in Refs. [a, b] (both focussed on other families of energy-based models, ConvNets in particular), are deeply related and consistent to what we observe in the RBMs: Machines trained with non-convergent MCMC chains (what we refer as out-of-equilibrium regime) and initialized from random configurations are very efficient for generating good samples, but cannot be used to approximate the unnormalized probability density of the dataset. Considering that both works predates our own, and are, to our knowledge, the first to identify this non-monotonous behavior, they will be necessarily discussed in the introduction, results and the conclusions of any future versions of our work, and the coincidence between convergent/non-convergent and equilibrium/non equilibrium phases will be explicitly mentioned. Furthermore, we believe that the observation of similar phenomena in significantly different energy based models make our results more general and solid.
> Yet, we want to stress that even if authors of [a,b] identified similar features in the MCMC sampling of EBMs trained using different recipes, we believe that our work provide a deep additional understanding of the underlying phenomena, mainly for 2 reasons:
>
> a) In our work, we identified a crucial parameter characterizing the problem: the MCMC mixing time associated with the model and its dependence on the number of parameter updates performed during the learning. Without this information, the separation between the non-convergent and convergent ML, depends in a complex and non-monotonous way on the dataset considered, the number of gradient updates, and the initialization of the negative chain. In our work, we show     that the equilibrium regime is achieved as long as $K$ is long enough when compared to the mixing time, a situation at which the initialization of the negative chain is totally irrelevant. This means that the number of steps $K$ needed to keep the machine working in the convergent ML regime, depends on how old the model is (for how long it was trained). This is so because the mixing time increases during the     learning. In other words, a learning scheme can work (learn the correct equilibrium model), up to a certain number of updates, but will eventually start to enter the non-convergent regime from a certain point - when the mixing time will be higher than the number of MC steps used for computing the negative term. And finally, the mixing time of the trained model is what limits how good the convergent ML model obtained with PCD can be (in the sense of how similar can the equilibrium configurations of that model to the real dataset ones) upon learning. If the mixing time remains small, a moderate value of $K$ is enough to train a good equilibrium model, while in the opposite case it can be quite costly. We observe that the model’s equilibrium states only improve with the number of training steps as long as $K$ is kept larger than the mixing time during training, and this is true no matter which recipe one follows. For instance, as it was also observed in [a], PCD-K trainings tend to train models that operate at the equilibrium regime (the MCMC sampling smoothly converges to the equilibrium measure). Yet, one can observe that the quality of these equilibrium samples is mostly limited by the value $K$ chosen, in the sense that quality stops improving with the number of parameter updates, at an age that depends on K (see Fig.5B and Fig.10 of SM).
>
> b) Our work characterizes in detail the strong connection between the dynamics followed during the learning and generating Gibbs sampling, and the number of MCMC steps needed to generate the best possible samples. We quantify the quality of the samples as a function of the sampling time, showing the existence of a clear best quality point that appears only when the negative term of the gradient is estimated with out-of-equilibrium data. Furthermore, we show that the position of this best quality point matches perfectly with the number of steps $K$ used to compute the negative term of the gradient, but only if the generating Gibbs sampling reproduces the same identical dynamics followed during the learning: that is, same initialization of the negative chain, and same dynamic rule for the     configuration updates. In the current manuscript, we just showed this match when using the random initialization (both in the training and the sampling) and Block updates for the visible and hidden nodes. If considered interesting, we can also show data showing the same effect if performing simulated annealing (always following the same annealing schedule) or mean-field updates, or by using alternative initialization such as starting from a uniform configuration of all ones or zeros. The main point is that when operating in the non-equilibrium regime, the RBM just learn to match as good as possible the dataset     after following the exactly same dynamics as the one used during the training process. A sampling using a different dynamics create very bad configurations.
>
> 2)** Energy of the configurations.** We have been tracking the energy since the beginning of the project as an additional quality estimator. We finally decided not to include it in the paper because the energy is normally a poor track for non-equilibrium behavior in disordered models since it evolves extremely slowly. This fact, can easily mislead you into believing that you have thermalized the system much sooner than what you have. Yet, after reading the referee’s comment, we agree with him/her that the evolution of the energy difference between the dataset and the generated samples would be useful to understand intuitively the difference between the equilibrium and the non-equilibrium phase, so we will include it as an additional estimator of the sample quality in the next version of the paper.
>
> 3) **More datasets.** Referee finds that our choice of datasets was a bit limited because we limited our results to binary pixels. As discussed previously with other referee pointing out the same comment, we can say that we have done experiments with CIFAR-10 32x32 with 3 color channels, and CelebA in 64x64 with 3 color channels, and multiple sequence alignments of protein families with 21 different amino acids+gaps, that display exactly the same phenomena. We are not allowed to include figures at this stage, but we can commit to include additional results on these datasets in a future version of the article if accepted.
>
> [b] Nijkamp, E., Hill, M., Zhu, S. C., & Wu, Y. N. (2019) arXiv:1904.09770.

---

> ### Author Response · Authors · 2021-08-09
> **Answer to ref g5Pg (2/2)**
>
> 4) **Strategies for equilibrating**
> The referee also requested some explanations about which are the right strategies for equilibrating. We agree with him/her that this is not clearly discussed in the manuscript and that the discussion could be improved. We will do it if we have the chance.
> The answer to that question depends on the actual goal of the training:
> “Numerically exact” convergent Training: In our case, we had in mind the application in which we want to learn an equilibrium model following the Gibbs distribution defined by the RBM’s energy function, that is also the best model to fit the dataset density. In this case, we seek a training set-up for which we can always progressively improve our model, in the sense that the equilibrium configurations are getting closer and closer to the training dataset, as we perform more and more training updates. For such an application, at least for MNIST, one needs to increase $K$ as the machine gets trained in order to keep the sampling always in equilibrium, no matter which MC initialization one considers. Of course, for such an application, an informative initialization of the negative chains (e.g. CD or even better, PCD), being nearer to the stationary state, allows using a lower $K$ than when using a random initialization. Still, in all these choices, the thermalization time will necessarily scale with the MCMC mixing time in the model, which is growing with the number of learning updates. In this setting, $K$ can become very large for long learnings, and hence, making the training very computationally demanding. In our case, we still showed that with the Rdm-10000, the machine remains at equilibrium at least until $69061$. The fact that the RBM enters an out-of-equilibrium regime for a larger number of updates only tells us that the equilibrating time has become higher than $K=10000$. It would still be possible to take even larger values of K to remain at equilibrium for a longer time if needed.
>
> Approximate (PCD) convergent training: a different answer would be given if we just speak of training models generating equilibrium configurations that are reasonably similar to the original training set, which might be enough for images that look reliable. For this purpose, the PCD-K approach is very well-suited since, as can be seen in our figures, it produces convergent models with a monotonous behavior. However, its use depends on the trade-off of computational budget/sample quality. While using PCD with $K~O(10)$ provides a convergent model, the quality of the samples is in general quite poor when compared to the ones obtained using Rdm-K with the same value of $K$. In fact, we can see on Fig. 10 (focusing on the indicator $\mathcal{E}^\mathrm{PSD}$ for instance) that the larger K is, the better are the samples generated by a machine trained with PCD. In fact, the samples obtained from an RBM trained with PCD are in general not as good as the ones obtained from the model trained with equilibrated negative chains (this is only true for the ages for which the mixing time has become larger than $K$, of course, otherwise both are the same). At the same time, for moderate values of $K\sim O(50)$, the sample quality provided by the Rdm-K method is much better than the one obtained when using PCD and the same $K$. Models trained in equilibrium do generate samples as good as the samples generated in the out-of-equilibrium scheme. Therefore, the good strategy clearly depends on the objective of the user, and its computational budget.
> We also want to stress that even PCD fails to fit convergent models for structured/clustered datasets (such as the genome one discussed in the SM), where the mixing times are enormous because the sampling gets trapped in metastable states. In such cases, we do     observe non-monotonous behavior in the quality of the generated samples too (training not shown in the current version).
> In summary, to our understanding, the best strategy to obtain a good equilibrium model would be to use PCD with a progressively increasing value of $K$. PCD-k training with fixed and low $K$ produces convergent models that are not able to generate good enough samples (at least as good as those produced with the out-of-equilibrium strategy for the same number of updates).

---

> > ### Comment · Reviewer_g5Pg · 2021-08-18
> > **Review Update**
> >
> > Thanks to the authors for the detailed response. Regarding previous work in [a], I find the response a) to be partially convincing but point b) not as convincing.
> >
> > Response to a): I agree that the careful analysis of mixing time is an informative experiment that provides additional insight beyond [a]. Although the metrics used by the authors seem to correspond with visual appearance, they are somewhat unconventional it might be more appropriate to use widely-used metrics for image appearance such as FID score to measure the quality of synthesized images compared to data images, but this is not a major objection.
> >
> > Response to b): These points seem to be largely covered by [a]. The authors of this work claim that out-of-equilibrium learning is more effective for image synthesis, and the same claim is made by [a]. The authors further claim that the RBM can learn synthesis from virtually any initialization distribution as long as it is consistent, and moreover is specifically tuned to that initialization, but this observation appears to be the motivation behind the noise-initialized learning (same as RMD in this work) introduced in [a].
> >
> > Equilibrium Strategies: This discussion seems reasonable to me. One other point to consider is the learning rate of the RBM. If the learning rate is very low, then persistent samples from a past checkpoint might effective act as samples from the current model, since the model changed very little. The authors of [a] find that both persistent learning and low learning rate are needed for equilibrium learning. While this situation might not be identical for RBM's, it might be interesting to consider how the rate at which the model changes interacts with the value of K needed for different learning outcomes.
> >
> > Overall: I am updating my score. The observations made by the authors have gone largely unnoticed for quite a long time and bringing more attention to this phenomenon is useful for the generative modeling community. The authors did a nice and thorough job of examining this problem.

---

> > > ### Author Response · Authors · 2021-08-30
> > > **Answer to the review update**
> > >
> > > We thank the referee for the positive comments, the suggestions, and for the update of the score.
> > >
> > > Concerning a), if the referee feels that it would improve the discussion, we can try to implement the FID score for a revised version of the paper.
> > >
> > > Concerning b), since we may not have explicitly written it in our first response, we reaffirm that future versions of this work will mention that the observation that best samples are generated when sampling the same number of MCMC steps K used to train the model, was previously discussed in [a] and [b]. Same for the fact that a long sampling of that model lead to fairly biased configurations. Yet, we still believe our work provides a more detailed and complementary study about this non equilibrium effect, mainly because it quantifies the effect on the sampling time (beyond visual inspection) in the generated configurations not only at K and long-samplings, but all the spectrum in between these two, and its dependence with the number of parameter updates used during the learning.
> > > Concerning the learned dynamics, we agree with the referee that the noise initialization learning proposed in [a] was justified precisely in repeating the same procedure during learning and sampling. Furthermore, we can also say that these works explicitly mention that the non-convergent short-run MCMC generator can be done using an arbitrary distribution. Yet, we believe that providing data testing this claim (as we were offering to include in a future version of our work), will be very illustrative for the community and will complement the results of [a] and [b]. With these tests, we mean that not only repeating the same initial condition is crucial, but also that the same sampling dynamics (eg: simulated annealing, mean-field learning or starting from uniform initial condition (all visible nodes to 0 for instance)). Otherwise, you observe a really poor generative power.
> > >
> > > Concerning the equilibrium strategy: we agree with the referee that a small learning rate should improve the “equilibration” when using a persistent chain, mainly because it produces an effect analogous to increasing K.  We didn’t perform such experiments in our work because we wanted to compare machines at different ages using different learning schemes and therefore with the same learning rate. In addition, this effect is specific for PCD. By using a smaller learning rate, the comparison in terms of number of updates would be harder, since a lower learning rate will increase the number of updates needed to reach good performance. Yet, if the referee thinks this would improve the paper, we could compare the quality of the samples of 2 models trained with PCD and the same K, with 2 distinct learning rates. We agree that lower learning rates will lead to better qualities for the same number of normalized number of updates.
> > > Finally, let us precise that if we agree that in many datasets, PCD is able to learn an equilibrated distribution, we have also observed that for others, very clustered, even PCD fails to learn a good equilibrated model and produces a non-convergent behavior. Concerning that point, we do not know if it is a general property of EBM or specific to the studied RBM.

---

### Official Review · Reviewer_35Jj · 2021-07-16

**Rating:** 6
**Confidence:** 3

**Summary:**

The paper discusses about some important factors affecting the results of RBMs.

1. The choice of mixing time.

2. The mixing time and the number of steps(k) affect two regimes of RBMs, namely equilibrium and out-of-equilibrium.

3. How to choose the number of steps(k). (A) short k for convincing samples in short learning times (B) large k for learning the correct equilibrium distribution of the RBM.

**Limitations And Societal Impact:**

Yes

**Main Review:**

1. The main problem of this paper is lack of innovation. The author compares the results of 3 different recipes (CD, PCD, Rdm) proposed by others, and puts forward that the number of MCMC steps has a dramatic effect on the model. This paper does not improve the previous algorithms a lot and not give a specific method or function concerning how to choose number of MCMC steps.

2. All the conclusions and results are proved empirically, not theoretically. However, the model has only been applied on two datasets, and deals with only binary nodes. The conclusions should be more convincing by training on more datasets.

3. The algorithm and results in section 2-5 can be presented in a clearer way by adding flow chart of the algorithm and tables of different results.

==============
After reading the response of authors, I update my comments (see below).

**Time Spent Reviewing:**

4

---

> ### Author Response · Authors · 2021-08-09
> **Answer to referee 35Jj**
>
> We take the chance to thank the referee for his/her interest in the paper and for the suggestions and comments. We believe they can help to improve the article and make it more useful for the community. We are going to address each of the 3 concerns raised by the referee below.
>
>   1. While we agree with the referee that this work does not strictly propose new algorithms, it does propose a precise and new recipe to use the existing ones in order to dramatically improve the generation power of RBMs in short training times. In particular, we show that a random initialization of the negative chains together with a fixed number K of Monte Carlo steps during the training can be exploited when resampling (generate new data from the learned model) using exactly the same dynamics with a potential short (and optimal) number K of MC steps. We show that this optimal number of sampling steps coincides precisely with K, the number of MCMC steps used to estimate the negative term of the gradient. We also show that, when using this Rdm-K recipe, at least above $K=50$ in MNIST (we only considered $K=10$ beyond that value), no further improvement in the quality of the new samples is observed upon increasing K (see Fig.5A of the SM). Furthermore, we also show that models trained with the Rdm-K recipe can generate better samples than those trained with PCD-K or CD-K at least for $K<1000$ (see Fig. 5B of the SM), and in fewer MC iterations. In particular, our results also highlight that PCD-K trainings obtain models that require a huge number of MCMC steps to generate new and uncorrelated samples, making them impractical for data generation purposes. In fact, while this effect is not too critical for MNIST, because the largest equilibrating times measured are always shorter than $10^5$ MCMC steps, it becomes impossible to generate new genomic data (see the second dataset in the SM) from PCD-k trained machines because it would require sampling times far beyond the typical time for a standard study (data not shown in the current version but could be discussed in a revised one). Yet, we also discuss that this speed-up associated with the Rdm-K recipe comes with the drawback of getting models that do not describe the good data density distribution, a goal for which, the PCD-k recipe is much better fitted. We propose to discuss these practical points in more detail in the main-text.
> Also, we want to stress that the match between K and the number of steps that lead to the best generating performance is not specific to the Glauber dynamics used in the text. When operating in the out-of-equilibrium regime, the RBM actually learns the dynamics used to train them. In fact, we can show experiments where we train the machine using simulated annealing or mean-field update dynamics to estimate the negative term of the gradient (always initializing from random configuration) and we reproduce exactly the same non-monotonous behavior in the quality of the generated samples but if and only if we use exactly the same dynamics to sample the equilibrium distribution of the model. The same situation arrives if negative chains are initialized for complete white or black images (all visible nodes to ones or zeros) during the learning and sampling. We could discuss these experiences if the referee considers them important.
> Finally, our work suggests at least a simple way to design a new algorithm for learning the correct model for the data density. Using the autocorrelation function, eq (10) of the main text, it is possible to choose the number of MC steps online (during the training) such that the autocorrelation is below a given threshold. In that way, the negative chains are guaranteed to have reached the equilibrium distribution and the equilibrium properties of the learned model will appropriately describe the dataset density,
> **In brief, our work shows how the scheme Rdm-K can be used for generating good samples in short training time and short generation time. This scheme can be generalized using other dynamics (even if not studied extensively here). Also, we can deduce from this work an automatic scheme to train a RBM always at equilibrium.**
>
>   2. We applied our method not in 2, but in 3 datasets of different nature (MNIST, a genome dataset and HQ-CelebA in black and white). If the referee considers that more datasets would make the results more solid, we can say that we also have results from CIFAR-10, and CelebA in 64x64 with 3 color channels and several protein families that describe exactly the same phenomena. We did not include them in the paper because we considered they were not adding much evidence to the discussion, but we could include them in an updated version of the paper. In particular, the memory effect discussed in the article does not depend on the particular architecture of RBM, but rather on the dynamics used during the learning to compute the negative term and the mixing time associated to this dynamics (in the presented cases, the dependence was mainly in the number K of MC steps used for sampling from random noise).
>
>   3. We stress that all suggestions aimed at improving the exposure of results are very welcome, and they will be taken into account for future versions of the paper.

---

> > ### Comment · Reviewer_35Jj · 2021-08-21
> > **review update**
> >
> > I carefully read all review comments and answers from authors, and agree with another reviewer that this paper provides a systematic experimental analysis on the popular learning methods of RBM. Some problems of the existing methods have not been noticed before, and the results of this paper will be helpful for further investigation in future. So I update my score.
> >
> > Some suggestions:
> >
> > 1) Can authors provide a reference protocol for training RBM in applications based on results of this paper? This would be helpful for readers.
> >
> > 2) For such an experimental research, there is no thorough theoretical, so the conclusions should be more convincing by results from CIFAR-10, CelebA or similar data sets as mentioned by authors.
> >
> > 3) Please describe the settings/techniques/quantities involved by experiments more clearly by adding flow charts and tables.

---

> > > ### Author Response · Authors · 2021-08-30
> > > **Answer to the review update**
> > >
> > > We wanted to thank the referee for his/her answer and his/her change of score. We want also to acknowledge that we will add results on these two dataset mentioned, and a visual description of the setting and experiment, as suggested.

---

### Official Review · Reviewer_zU2d · 2021-07-19

**Rating:** 7
**Confidence:** 5

**Summary:**


The authors extensively analyze the training of Restricted Boltzmann Machines (RBMs).
They show that there is definite room for improvement in training RBMs in terms of

- *Evaluation:* The authors show that despite its popularity, log likelihood is not the only metric which should be tracked while training the RBM and show that bad machines can have a deceptively good log likelihood but not be able to generate good samples and vice versa.
- *Methods:* Despite their popularity, both CD-k and PCD-k can be outperformed in terms of the quality of generated samples by Rdm-k which uses chains started at uniformly sampled states to compute the negative statistics

Moreover they empirically show that for Rdm-k, smaller values of k lead to models which memorize data (Figure 2A $\Delta S$) and a larger k leads to generalization (likelihood of generated samples is close to that of the data) (Figure 2A $\mathcal{L}$).

**Limitations And Societal Impact:**

Adequately addressed

**Main Review:**

### Strengths

- The authors conducted extensive experiments to analyze what had been thought to be an un-interesting and known aspect of RBM training and have demonstrated surprising results.
- The entire paper (especially the introduction) is well written, motivation is well presented and the experimental evaluation is well explained.
- The paper proposes and summarizes multiple very useful metrics which can be used by practitioners to track their RBM training.

### Weaknesses

- [Clarified in rebuttal] The evaluation and the methods proposed, while highly informative and useful, are computationally expensive.
  - The authors mention this in their supplementary material.
- While the mixing time estimation is interesting a comparison with theoretical bounds, in particular [1] would have been more interesting.
- Possible citations:
  - [2] could be cited as a more recent survey of training methods along with the citation on line 33
  - [3] was recently published in ICML and proposes a novel method to train RBMs which is unrelated to CD-k, a similar analysis of that method compared to Rdm-k would be very interesting.
  - [4] proposed a method to train RBMs using random walk tours which provide an almost unbiased estimate of the negative statistics
    - they additionally observed a similar effect where samples from shorter tours (k) demonstrated memorization and longer tours (k) demonstrated generalization, which is corroborated by the observations in the current paper.
  - [5] proposed using the up and coming unbiased couplings method to train EBMs including RBMs, since this method is scalable to larger RBMs it could be the subject of such a study.

### Other criteria

- Originality : Existing problem, marginally novel solution, exciting and novel results.
- Quality : Extremely high quality of writing.
- Clarity : Experiments are well justified,  motivations are well explained, results are well elucidated.
- Significance : High significance in terms of insight, prohibitive computational cost will hamper utility.

[1] - Tosh, Christopher. "Mixing rates for the alternating Gibbs sampler over restricted Boltzmann machines and friends." International Conference on Machine Learning. PMLR, 2016.

[2] - Song, Yang, and Diederik P. Kingma. "How to train your energy-based models." arXiv preprint arXiv:2101.03288 (2021).

[3] - Grathwohl, Will, et al. "Oops I Took A Gradient: Scalable Sampling for Discrete Distributions." arXiv preprint arXiv:2102.04509 (2021).

[4] - Savarese, Pedro, Mayank Kakodkar, and Bruno Ribeiro. "From monte carlo to las vegas: Improving restricted boltzmann machine training through stopping sets." Proceedings of the AAAI Conference on Artificial Intelligence. Vol. 32. No. 1. 2018.

[5] - Qiu, Yixuan, Lingsong Zhang, and Xiao Wang. "Unbiased contrastive divergence algorithm for training energy-based latent variable models." International Conference on Learning Representations. 2019.

**Time Spent Reviewing:**

4

---

> ### Author Response · Authors · 2021-08-09
> **Answer to referee zU2d**
>
> We take the chance to thank the referee for his/her interest on the paper, the positive comments and the useful suggestions. We try to respond to each of the 3 concerns mentioned by the referee.
>
> Before entering into the weaknesses, we would like to highlight that, Fig 2A shows that the Rdm-K recipe, is a pretty good method to generate samples even for short K, and that the longer the training, the higher the quality of the samples. In fact, even at $K=10$, all the indicators show very good performance when taking samples generated using the same dynamics as for the training and stopping at $K=10$. The discrepancy between the log-likelihood computed over the dataset and generated data for long sampling time show that the learned RBM is not a good model for the dataset-density.
>
> Concerning the list of weakness:
>   1. **Computational cost**. It is not clear for us to what precise computation the referee is referring to here. Concerning the learning method, the complexity of Rdm/PCD/CD-K clearly depends on how many steps $K$ are used. While in many works $K\sim \mathcal{O}(10)$ is used, we show that it is in general better to use PCD-K with $k\sim 100$ to reach quasi-equilibrated samples at each update and therefore learn a model with good equilibrium properties, or Rdm-K with $K\ge 50$ if taking advantage of the memory effects of the learning for this scheme. We further show results for $k=10000$ in order to illustrate, that it is, in principle, possible to obtain equilibrated samples during almost all the training time even using a completely random start, and that models always trained at equilibrium describe better the dataset density than those trained using short values of $K$s. We agree that such high values of Ks are impractical, but we believe that these experiments are useful to prove our point: the mixing time of the MC chains are crucial for learning the correct model and to generate new data as equilibrium configurations. In fact, we believe that the correct scheme would be to increase K progressively during the learning, large Ks are only necessary when willing to reach a very large number of parameter updates. It was also interesting to compare the mixing time between the PCD-k approach and the Rdm-K approach.
> On the proposed evaluations methods. All the proposed metrics are not computationally expensive. Most of them depend on generated samples at various MC steps, which is quite common for an energy-based-model. The generation cost is $O(N_v \times N_h \times M \times t)$, where $N_v$ is the number of visible nodes, $N_h$ of hidden nodes, $M$ the number of generated data and $t$ the number of MC steps. Concerning the different methods, the computation of the log-likelihood is probably the most demanding one, since it imposes an annealing over many temperatures and is prone to higher uncertainties. The 3-point correlation is clearly prohibitive in practice. However, the 2-point correlation and the ATS scores have reasonable complexities which make them good candidates for online evaluation, and  we show they are as informative as the other metrics. For the former, having $M$ samples, the cost is dominated by $O(N_v \times M \times \min(M,Nv))$ which is of the order of magnitude of the generation of new samples for one MC step if $N_v \sim N_h \sim M$. The latter is also dominated by the same complexity since we need to compute the distance between all the samples (true ones and generated ones). We therefore believe that these two metrics are of practical use during the learning and not too computationally expensive.
>
>   2. **On the mixing times**. We thank the referee for letting us know about Ref. [1] concerning the mixing rates for the alternating Gibbs sampler, which is exactly the case of the RBM. In particular, if our paper is accepted, we will include this reference as a related work concerning rigorous results for the mixing rates. In [1], two analytical results concern the binary-binary RBM. In the first one, it is shown that the mixing time is upper-bounded when $|| W ||_1 || W^T ||_1 < 4$ where $||.||_1$ is the L1 norm for rectangular matrices. This rigorous result seems linked to the potential presence of a second-order phase transition in the RBM during the learning. In the ref. [a], the authors show that under reasonable hypotheses, the RBM undergoes a second-order phase transition when the highest eigenvalue of the weight matrix W overpasses a given threshold. For binary-sigmoid RBM, this threshold is 4 (which is not directly related to the threshold value in [1]). During the learning, the RBM goes from a paramagnetic phase, where the mixing time is very short, towards a condensation phase where it is much higher. In particular, at the point of the transition, the mixing time diverges, as is known from the theory of critical phenomena. In fact, for the MNIST dataset, we did measure the mixing time using eq. (10) of the main text. What we observed is that before the first eigenvalue passes the threshold 4, the mixing time is about $O(1)$ MC steps, then it diverges for a very short time at the transition, and goes back to $O(100)$ after it. Therefore, the theoretical bound of ref. [1], is valid only until a bit before (during the learning) than the threshold of ref. [a], marking the point where the machine begins to learn features of the data. This means that the upper bound of ref. [1] refers only to the uninteresting regime, when the  machine has not yet learned anything. The second result concerning the RBM in Ref. [1], is a lower bound on the mixing time. However, this result only states that it is possible to find an RBM with a well-designed weight matrix (and no bias), such that the mixing time is lower-bounded. In practical cases, this result can not be used for trained RBMs because nobody tells you that this matrix has any connection with that of a properly trained RBM.
>
>   3. **References**. We thank the reviewer for useful suggestions of citations. The ref [2] can clearly be cited as a review of training methods, more focused on EBM. Concerning refs [3,4,5], they all propose interesting new methods to improve the MC sampling. In [3] a scheme of variable selections is introduced to select the variables during the MC process. In [4], a new criterion is used to adjust the number of MCMC steps used to compute the negative term of the gradient, that potentially reduces the number of MC steps done during the learning. In [5], a new estimator of the negative term is used, together with an elaborated stochastic process for which the number of sampling MCMC steps can vary. All these approaches potentially reduce the mixing rate of the chains, and we agree that we should cite them as improved MC methods. In any case, none of these improvements change the general conclusion of our work: if working in an out-of-equilibrium regime (having done a number of MCMC steps not long enough), the RBM will not be trained properly, and the quality of the samples obtained during the sampling process will be non-monotonic. Let us add that, using the autocorrelation function Eq (10), we can design a Rdm-k algorithm with $k$ that varies in a way that we can ensure equilibrium for the computation of the gradient all over the training. In such a setting, we have the guarantee that the trained machine will correctly represent the dataset-density.
>
>
>
>
> [a]“Thermodynamics of Restricted Boltzmann Machines and Related Learning Dynamics”, A. Decelle, G. Fissore and C. Furtlehner, Journal of Statistical Physics volume 172, pages 1576–1608 (2018)

---

> > ### Comment · Reviewer_zU2d · 2021-09-03
> > **Thanks for addressing my concerns.**
> >
> > I thank the authors for addressing my concerns and for taking my suggestions into account.
> > Regarding the computational cost, I was indeed referring to the evaluation and not the Rdm-k method which is as expensive as CD-k.
> > I apologize for any confusion this may have caused.
> >
> > While I maintain my score, I would like to upgrade my confidence in accepting this paper based on the other reviews and author responses.
> > I would also like to add that the other two weaknesses I had listed have been adequately addressed by the authors as well.

---

### Decision · Program_Chairs · 2021-09-27

**Decision:**

Accept (Poster)

**Comment:**

This experimental paper has been well received by the reviewers, who are overall fairly confident that the paper deserves to be accepted. They also unanimously point out the quality of writing and presentation. The reviewers have done a great job, and I trust the authors to implement the promised revisions before publication. The main weakness appears to be a lack of originality. But given the timeliness of the models studied here and the overall quality of presentation, this papers complements well existing literature pointing at the non-equilibrium properties of energy based generative models. Moreover the reviewers acknowledge mostly convincing responses to their comments. In particular, I think it is important that, as promised by the authors, they implement the numerical experiments on more datasets, especially given the pure experimental nature of the paper. Second, a reviewer commented: "Can authors provide a reference protocol for training RBM in applications based on results of this paper? This would be helpful for readers." I completely back-up that point. I think that this would add to the paper. Last but not least: it is of crucial importance that the authors extensively discuss and compare their results to reference https://arxiv.org/abs/1903.12370 , and explain in details how their contributions complements it and confirms further its initial findings in a different context.